# Model selection may not be a mandatory step for phylogeny reconstruction

Shiran Abadi [1], Dana Azouri[1,2], Tal Pupko[2] & Itay Mayrose [1]

Determining the most suitable model for phylogeny reconstruction constitutes a fundamental step in numerous evolutionary studies. Over the years, various criteria for model selection have been proposed, leading to debate over which criterion is preferable. However, the necessity of this procedure has not been questioned to date. Here, we demonstrate that although incongruency regarding the selected model is frequent over empirical and simulated data, all criteria lead to very similar inferences. When topologies and ancestral sequence reconstruction are the desired output, choosing one criterion over another is not crucial. Moreover, skipping model selection and using instead the most parameter-rich model, GTR+I +G, leads to similar inferences, thus rendering this time-consuming step nonessential, at least under current strategies of model selection.

[1] School of Plant Sciences and Food Security, Tel Aviv University, Ramat Aviv, Tel-Aviv 69978, Israel. [2] School of Molecular Cell Biology & Biotechnology, Tel Aviv University, Ramat Aviv, Tel-Aviv 69978, Israel. These authors contributed equally: Shiran Abadi, Dana Azouri. These authors jointly supervised this work: Tal Pupko, Itay Mayrose. Correspondence and requests for materials should be addressed to T.P. (email: talp@tauex.tau.ac.il) or to I.M. (email: itaymay@tauex.tau.ac.il)

Probabilistic evolutionary models form the basis of sequence data analyses. Parameter inference, whether performed within the maximum likelihood (ML) or Bayesian inference paradigms, relies on explicit definition of the substitution process, which may vary in spatial manner (across the alignment sites) and in temporal manner (branches of the phylogeny). Over the last 50 years, a plethora of evolutionary models has been developed, each relying on a different set of assumptions regarding the dynamics of nucleotide evolution. Such assumptions, quantified by several parameters, determine whether the substitution rates between all pairs of nucleotides are identical or independent, whether the stationary frequencies of the nucleotides within the analyzed data are equal or allowed to vary, whether a proportion of the sites are fully conserved, and whether heterogeneous rates of evolution are allowed across the alignment sites. Altogether, these produce varied alternatives that account for different processes of evolution[1–8].

Accounting for more parameters grants a model the flexibility to fit different datasets and capture their complexity. However, the expected error of each estimate increases with the increase in the number of parameters, which is problematic mainly when data are scarce. Selecting the most suitable model for describing the evolutionary process has been addressed under both the frequentist and Bayesian approaches, by proposing statistical criteria to compare the fit of competing models. Under the frequentist approach, the fit of the data to each substitution model, together with the model parameters, tree topology, and branch lengths is assessed through iterative optimizations of the likelihood function. The estimated ML scores are then compared through one of several possible criteria. For example, the hierarchical and dynamic likelihood ratio tests (hLRT and dLRT, respectively) criteria perform a sequence of likelihood ratio tests between pairs of nested models, until a model that cannot be rejected is reached. While in hLRT the order in which parameters are added is defined a priori, in dLRT all models that differ in one parameter are compared in parallel and the hierarchy proceeds with the model that maximizes the log-likelihood difference. Thus, dLRT enables a different order of hypotheses testing for different datasets[9]. Other criteria compute the ML for all the candidate models, but assign different penalties according to the data size or the number of parameters included in the model. The most commonly used criteria are the Akaike information criterion (AIC)[10], the corrected AIC (AICc)[11], the Bayesian information criterion (BIC)[12], and the decision-theory criterion (DT)[13] (summarized in Table 1).

Notably, handling the uncertainty within model testing by the ML criteria depicted above is accomplished by accounting for the number of parameters assessed in the computation, but not for the type of processes they represent. For example, the penalty for a parameter that distinguishes between transition and transversion would be identical to the penalty imposed for a parameter that assesses the number of invariant sites. In contrast, under the Bayesian approach, model selection can be performed using the marginal likelihood, which is the probability of the data given the model, while marginalizing the estimates (Table 1). The magnitude of the Bayes factor (BF), namely, the ratio of the marginal likelihoods of two models, quantifies the strength of evidence that one model is more appropriate to describe the data than the other[14]. Since the marginal likelihood for phylogenetic interpretation consists of high dimensionality and the wide range of values cannot be enumerated, its computation is not always feasible. Several methods that estimate the Bayes factor or the marginal likelihood for model selection in phylogenetic analyses have been proposed, with variable tradeoff between computation times and accuracy[15–20].

Obviously, no evolutionary model can fully capture the genuine complexity of the evolutionary process, such that even the most adequate one merely provides an approximation of reality[21]. Nevertheless, it has been claimed that using an inadequate model may result in erroneous phylogeny[22,23], and thus model selection is considered an integral part of the phylogenetic reconstruction procedure. However, the use of different criteria often leads to the selection of different models. Several studies used simulations to evaluate the performance of different model selection criteria, focusing on the ML criteria[9,13,24–30]. These studies established that AIC is more permissive, tending toward more complex models, while BIC and DT exhibit the opposite pattern, tending to choose simpler models. Nevertheless, as far as accuracy of choosing the generating model is concerned, there appears no consensus regarding the preferred criterion. Posada[30] and Posada and Crandall[9] initially concluded that methods that rely on likelihood ratio tests perform better than AIC and BIC. However, a later study by Posada and Buckley concluded that the use of hLRT may not be effective for real data and therefore averaging different models according to the weights given by AIC or BIC is preferred[29]. Increasing this ambiguity, an additional study showed that BIC and DT select the generating model more frequently than AIC and hLRT[24], whereas under other simulation conditions, AIC was shown to be more accurate than BIC[26]. Notably, these studies did not thoroughly examine the various tasks that are downstream to model selection. Hence, it is unclear whether the use of alternative best-fitted models according to different criteria would result in different inferences. It was argued that the inferred topology should be quite robust to the selected model[25–28,31], yet other applications, such as branch lengths estimation and ancestral sequence reconstruction, may be more sensitive[13,27,28,30,32].

## Table 1 Model selection criteria procedures

| Criterion | Procedure |
|---|---|
| AIC | ML is computed for every candidate model and the model with minimal $\{-2\ell + 2K\}$ is selected |
| AICc | Based on AIC but penalizes also for the data size. Namely, the model with minimal $\{AIC + \frac{2K(K+1)}{n-K-1}\}$ is selected; advised to be used instead of AIC when $\frac{n}{K} < 40$[29] |
| BIC | ML is computed for every candidate model and the model with minimal $\{-2\ell + K \ln n\}$ is selected |
| DT | Based on BIC but incorporates relative branch-length error as a performance measure |
| hLRT/dLRT | Sequential likelihood ratio tests between pairs of nested models until one cannot be rejected. Topologies are fixed to allow nesting. While in hLRT the order in which parameters are added is defined a priori, in dLRT all models that differ in one parameter are compared in parallel and the hierarchy proceeds with the model that maximizes the log-likelihood difference. Thus, dLRT enables a different order of hypotheses testing for different datasets |
| BF | The ratio between the marginal likelihood of two models. A ratio above 10 implies strong support for the model at the numerator |

$\ell$, the maximum log-likelihood of model $M$ is computed as $\log P(D|M, \theta)$; $D$, $M$, $\theta$ represent the data, the model, and the parameters estimates (i.e., the model parameters, branch lengths, and tree topology), respectively. $K$ is the number of parameters; $n$ is the data size (usually defined as the number of sites in the alignment[33]). The marginal likelihood is computed as $P(D|M) = \int P(D|M, \theta) P(\theta|M) d\theta$

In this study, we first present a literature survey that demonstrates the lack of consensus regarding which model selection criterion should be used for phylogenetic studies. A possible reason for the inconsistency in the conclusions of the above-mentioned studies is that they were established on few batches of datasets that were simulated under combinations of delimited values of number of taxa, alignment length, and sequence divergence. To avoid potential biases in our in-depth analysis of the performance of alternative criteria, we assembled an extensive collection of empirical datasets that ranges over a wide variety of biological parameters. We show that alternative criteria yield similar downstream phylogenetic inferences and that simply using the most complex nucleotide substitution model, GTR+I+G, leads to similar results, at least for the inference of tree topology and ancestral sequences.

## Results

**Current model selection practices.** We examined current practices applied by the community by sampling 300 phylogenetic studies that used jModelTest[33,34] for model selection during 2017–2018 (the data are available in Supplementary Data 1). This survey revealed that the criterion of choice is quite arbitrary: 41%, 21%, and 5% of the researchers opted to use the model selected by AIC, BIC, and AICc, respectively, while DT was used in a single study only. Notably, 36% of the studies did not specify the used criterion, and 4% declared that the same model was selected by more than one criterion. Thus, despite the wide-use of model selection procedures in phylogenetic studies, it seems that either the criterion used is not regarded as a crucial consideration, or that a thorough research to guide the community on the merits of the criteria is still—surprisingly—missing.

**Effect of model selection on phylogeny reconstruction tasks.** We sought to understand the relative merits of the various model selection criteria by comparing their performances on phylogeny reconstruction. The datasets that were used for this examination (referred to as simulation set $c_0$) were simulated based on varied realistic data conditions, derived from three empirical databases: PlantDB[35], Selectome[36], and PANDIT[37] (7200 datasets altogether), each with one of 24 commonly used substitution models (see Methods). To assess the effect on the resulting tree, we reconstructed the corresponding tree using the selected model by each of the six criteria: AIC, AICc, BIC, DT, dLRT, and BF, and compared each selected tree (reconstructed with the selected model of each criterion) to the true tree (that was used to simulate the sequence data). First, when reconstruction of the true topology was examined, the six criteria performed similarly as they all selected models that correctly recovered the topology of the true tree in 50–51% of the datasets (Table 2, first column). Second, we investigated which criterion tended toward models that yielded topologies with the minimal distances from the true trees by computing the Robinson-Foulds (RF) distance[38]. Then, we ranked the criteria from lowest distance (rank 1) to the largest. Our results demonstrated that all criteria performed very similarly ($p$-value > 0.05 for all pairs of criteria; pairwise Wilcoxon signed rank tests adjusted for ties with the Bonferroni correction for multiple testing). In fact, the averages of the criteria ranking across all datasets were highly similar (Table 3, first column). Furthermore, each pair of criteria agreed on the topologies of more than 83% of the 7200 datasets (Fig. 1a). These results suggest that the choice of the model selection criterion has marginal impact on the resulting tree topology. Third, we analyzed the accuracy of the inferred branch lengths of the reconstructed trees by measuring their distances from the true trees using the branch score (BS) distance[39], and ranked them

### Table 2 Percentage of accurate topologies

| Strategy/ simulation set | $c_0$ | $c_1$ | $c_2$ | $c_3$ |
|---|---|---|---|---|
| AIC | 50.51 | 50.44 | 50.64 | 36.50 |
| AICc | 50.51 | 50.47 | 50.58 | 36.60 |
| BIC | 50.44 | 50.47 | 50.69 | 35.80 |
| DT | 50.47 | 50.44 | 50.68 | 35.70 |
| dLRT | 50.29 | 50.26 | 50.78 | 35.50 |
| BF | 50.62 | | | |
| GTR+I+G | 50.82 | 50.94 | 51.11 | 36.40 |
| JC | 48.31 | 48.81 | 50.33 | 35.40 |
| True model | 50.17 | | | |

The table presents the percentages of correctly inferred topologies of every simulation set by each reconstruction strategy. The top six rows represent the accuracy obtained by reconstruction with the models selected by the various model selection criteria. The next two rows represent the reconstructions of the GTR+I+G and JC models regardless of model selection. The true model represents reconstruction with the model used to simulate each dataset, and therefore is applicable to simulation set $c_0$ only. BF criterion was not run for the complex simulation sets $c_1$–$c_3$ (see Methods). The percentages of simulation set $c_3$ were computed over a subset of 1000 datasets that represent coding sequences

### Table 3 Mean strategies ranking according to topological distances

| Strategy/simulation set | $c_0$ | $c_1$ | $c_2$ | $c_3$ |
|---|---|---|---|---|
| AIC | 4.96 | 3.97 | 3.99 | 3.86 |
| AICc | 4.97 | 3.97 | 4.00 | 3.88 |
| BIC | 4.97 | 3.98 | 4.00 | 3.96 |
| DT | 4.97 | 3.98 | 4.00 | 3.96 |
| dLRT | 4.98 | 4.01 | 4.01 | 4.03 |
| BF | 4.98 | | | |
| GTR+I+G | 4.97 | 3.92 | 3.91 | 3.91 |
| JC | 5.20 | 4.16 | 4.10 | 4.40 |
| True model | 4.99 | | | |

For each dataset, the ranking of the strategies was determined according to the Robinson-Foulds distance from least (ranked as 1) to most distant. The ranks range from 1 to 9 for set $c_0$ and from 1 to 7 for sets $c_1$–$c_3$. The top six rows represent the accuracy obtained by reconstruction with the models selected by the various model selection criteria. The next two rows represent the reconstructions of the GTR+I+G and JC models regardless of model selection. The true model represents reconstruction with the model used to simulate each dataset, and therefore is applicable to simulation set $c_0$ only. BF criterion was not run for the complex simulation sets $c_1$–$c_3$ (see Methods). The ranks of simulation set $c_3$ were computed over a subset of 1000 datasets that represent coding sequences

in a similar manner as for the topological distances. Evidently, the averages across the rankings of all datasets were similar for all criteria (Table 4; first column). Still, the ranks of BIC and DT were statistically significantly lower (i.e., more accurate) than AIC and BF ($p$-value < 0.05; pairwise Wilcoxon signed rank tests).

Notably, it is possible that the average over all datasets conceals superiority of some criteria in certain ranges of the data. In addition, ranking the criteria according to their performances allows comparison across different tree sizes, but does not reflect the magnitude of the errors. To address these limitations we binned the datasets according to the tree size, i.e., the number of tree nodes for the analysis of topological distances, and the total branch lengths (TBL) for the analysis of branch-length distances. Then, we compared the actual distances of the datasets within each bin between the criteria, rather than their ranks. Still, no apparent differences were observed between the averages obtained by the various criteria, both for topology and branch-length estimates across increasing tree sizes ($p$-value > 0.05 for all pairs of criteria for all bins; paired $t$-tests adjusted for ties following the Bonferroni

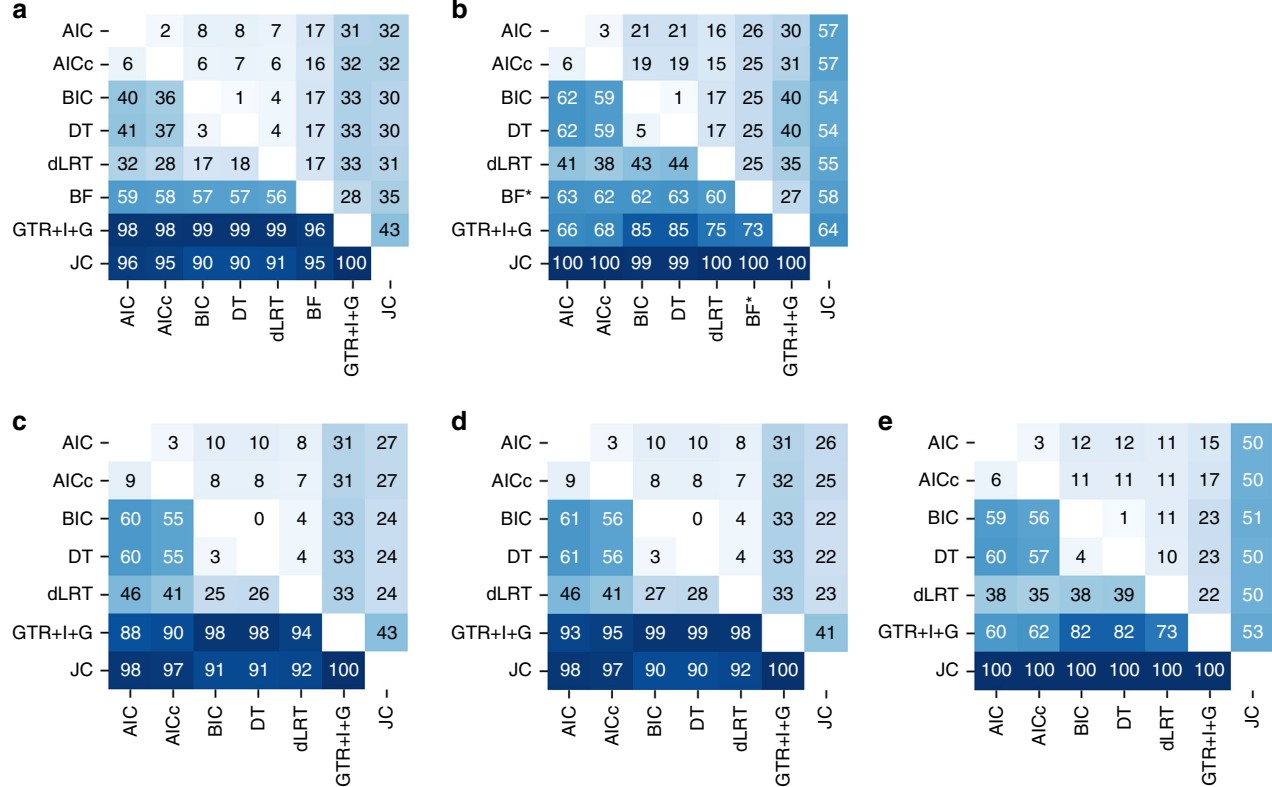

**Fig. 1** Pairwise incongruencies on the trees inferred by the evaluated strategies. The number within each cell represents the percentage of discrepancies between the two strategies at the row and column. The best-fitted model was computed for each criterion, and the trees were reconstructed using ML optimizations according to this model, as well as for the most complex and simplest models—GTR+I+G and JC. For each pair of strategies (rows and columns) the percentage of non-identical trees over 7200 datasets is presented (see * and ** below). The upper right triangles represent the percentages of different topologies and the lower left triangles represent different branch-length estimates. Clearly, two different models lead to different branch-length estimates, hence the latter reflect the percentages of differently selected models. The panels represent the following datasets: **a** simulation set $c_0$, **b** the empirical set, **c** simulation set $c_1$, **d** simulation set $c_2$, and **e** simulation set $c_3$**. (*) The percentages in the row and column of the BF criterion in panel b were computed over a subset of 1500 empirical datasets for which BF was computed (marked with an asterisk; see Methods). The analysis over this subset of 1500 datasets for all comparisons is presented in Supplementary Figure 1. (**) The percentages of the simulation set $c_3$ were computed over a subset of 1000 datasets that represent coding sequences (see Methods)

correction for multiple testing; Supplementary Figures 2a and 3a, Supplementary Data 2, 3).

Subsequently, we examined whether the use of different model selection criteria has an effect on ancestral sequence reconstruction, as an example of an analysis which is downstream to phylogeny inference. To this end, each of the selected models (together with their corresponding selected trees) was used to infer the root sequence for 1000 datasets (see Methods). Then, we measured the percentage of incorrectly inferred sites of each inferred sequence compared to the corresponding true sequence. The different criteria produced highly similar results. Namely, for all criteria, in 44% of the datasets the inferred sequence was identical to the true one, and in 97% of the datasets fewer than 5% of the sites were erroneous. Even though the sequence divergences in these simulated datasets reflect those found in many empirical datasets, we hypothesized that noticeable differences would become apparent when more divergent sequences, representing more challenging inference cases, were simulated. To this end, we resized all trees to several scales and repeated the analysis. For all criteria, the average percentage of incorrectly inferred sites increased with the increase in sequence divergence, however, the dissimilarities between every pair of the inferred sequences were still negligible (Fig. 2a and Supplementary Table 1). This suggests that choosing among model selection criteria has minor effect on the accuracy of ancestral sequence reconstruction.

**Performance of model selection under model misspecifications.** Evidently, the results presented above were derived from data that were simulated using the same set of models that were available for inference. Moreover, these evolutionary models are clearly an oversimplification of realistic sequence dynamics. This is reflected by the lower percentages of non-similar phylogenies, i.e., of incongruencies, between pairs of criteria over the simulated datasets compared to the empirical ones (Fig. 1a, b). For example, only 8% of the topologies were inferred differently by AIC and BIC over the simulated datasets compared to 21% over the empirical datasets. Consequently, such simulations may not be as challenging as real data analysis and may diminish the differences among model selection criteria.

To examine the effect of the inevitably simplifying assumptions of the simulations, we augmented their complexity by generating data with models that are not as simplistic as those available for model selection and phylogeny inference. To this end, we integrated two layers of complexities within two additional simulation sets. The datasets of the first set, $c_1$, were simulated using across-site variation of the substitution model; the datasets of $c_2$ were simulated as those of $c_1$ but also with rate heterogeneity across sites inferred from the empirical datasets, rather than sampled from the gamma distribution. The rates that were used to simulate $c_1$ and $c_2$ were inferred from the empirical datasets. As before, tree reconstructions of these two complex simulation sets yielded only few distinct topologies: <10% of the 7200 simulated

datasets resulted in different topologies between each pair of model selection criteria (Fig. 1c, d). In addition, all model selection criteria were essentially identically accurate in the topology inference ($p$-value > 0.05; pairwise Wilcoxon signed rank tests between all pairs following the Bonferroni correction for multiple testing; Tables 2, 3; Supplementary Figures 2b and 2c; Supplementary Data 2). Very small differences were also observed between the branch-length estimates of the model selection criteria (Table 4; Supplementary Figure 3b and c; Supplementary Data 3). We next examined the ancestral sequence reconstruction. Interestingly, while the distances between the reconstructed sequences and the true sequences were larger for the most complex simulation set ($c_2$) compared to the former analysis (reaching average distances of 0.34 compared to 0.25), the distances between each pair of criteria remained negligible

**Table 4 Mean strategies ranking according to branch-length distances**

| Strategy/simulation set | $c_0$ | $c_1$ | $c_2$ |
|---|---|---|---|
| AIC | 4.79 | 3.93 | 4.04 |
| AICc | 4.76 | 3.93 | 4.02 |
| BIC | 4.68 | 3.81 | 3.83 |
| DT | 4.67 | 3.80 | 3.83 |
| dLRT | 4.69 | 3.86 | 3.89 |
| BF | 4.77 | | |
| GTR+I+G | 5.50 | 4.22 | 4.39 |
| JC | 6.46 | 4.45 | 4.01 |
| True model | 4.68 | | |

For each dataset, the ranking of the strategies was determined according to the branch-length distance from least (ranked as 1) to most distant. The ranks range from 1 to 9 for set $c_0$ and from 1 to 7 for sets $c_1$ and $c_2$. The top six rows represent the accuracy obtained by reconstruction with the models selected by the various model selection criteria. The next two rows represent the reconstructions of the GTR+I+G and JC models regardless of model selection. The true model represents reconstruction with the model used to simulate each dataset, and therefore is applicable to simulation set $c_0$ only. BF criterion was not run for the complex simulation sets $c_1$ and $c_2$. Note that the analysis of branch-length distances was not performed for simulation set $c_3$ since the branch-length estimates are not comparable between the true and reconstructed trees, i.e., in the latter these represent substitutions per nucleotide site and in the former they represent substitutions per codon site

(Fig. 2b). Taken together, these analyses suggest that there is no clear preference of one model selection criterion over another, both for the phylogenetic inference and for ancestral sequence reconstruction.

In spite of the enhanced complexity, the simulation sets presented above were still generated based on the homogeneous nucleotide substitution models used for inference. In order to examine whether analyses over data that were generated based on other evolutionary patterns are in line with the deduction above, we used a codon model, M8[40], to simulate an additional set termed $c_3$. The rates that were used to simulate these datasets were inferred from a subset of 1000 alignments of coding genes included in the empirical set. The succeeding analyses over these generated codon alignments were performed using the nucleotide substitution models, similar to $c_0$–$c_2$. As before, the percentages of accurate topologies obtained by all criteria were highly similar (Table 2, last column; it should be noted that these percentages were lower than those of simulation sets $c_0$–$c_2$, probably due to the increased complexity). Likewise, the incongruencies over the reconstructed topologies were minor, similar to the previous analyses (Fig. 1e). Since branch lengths represent substitutions per nucleotide site in the reconstructed trees rather than substitutions per codon site as in the true tree, the distances between them were not measured for simulation set $c_3$.

**Model selection criteria incongruency**. Evidently, the various model selection criteria appeared to perform similarly in all of the presented analyses. One possible explanation to support these findings is that all the criteria tend to select the same substitution model and thus the reconstructed phylogenies are identical. To examine this hypothesis, we assessed the extent of discrepancies among the various model selection criteria over the empirical datasets. Overall, low agreement was observed among the various criteria (Fig. 3a, lower triangle). The selections of AIC and BIC were frequently different (in 62% of the empirical datasets). In comparison, higher agreement was observed between dLRT with either AIC or BIC (ca. 40% disagreement with both), while BF had a very low agreement with all other criteria (60–63% disagreement). As expected, high agreement percentages were

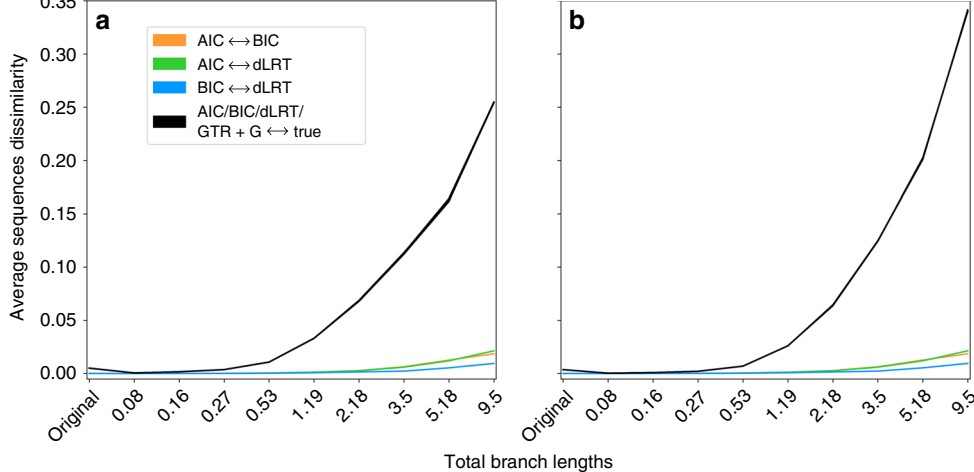

**Fig. 2** The impact of model selection criteria on ancestral sequence reconstruction. The $y$ axis represents the fraction of sequence sites that were different between every pair of root sequences, averaged across 1000 examined datasets: the black curves (which merge due to negligible differences) represent the comparison between the true root sequence and the inferred one according to the models selected by each of the criteria AIC, BIC, and dLRT, or consistently using GTR+G, and the colored curves represent the differences between every pair of criteria. The results of AICc and DT were similar to AIC and BIC, respectively, thus they are not shown. To increase the variety of sequence divergence, the analysis was repeated for trees that were resized to several scales ($x$ axis). The left and right plots represent the analysis on **a** simulation set $c_0$; **b** complex simulation set $c_2$. For the numerical estimates, see Supplementary Table 1

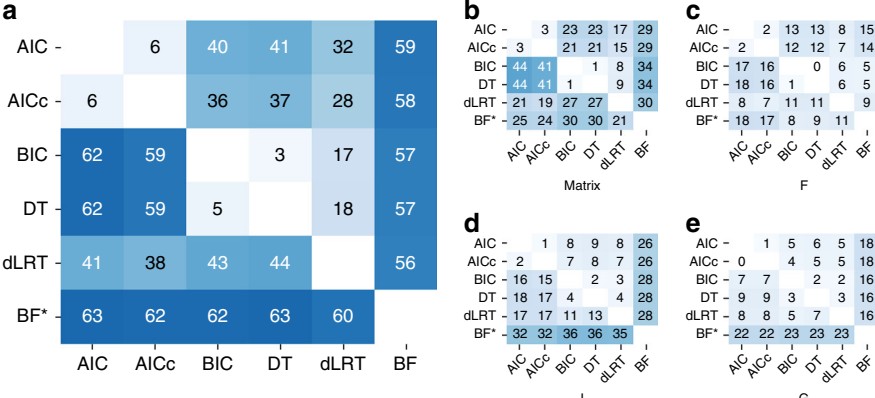

**Fig. 3** Incongruency over the selection of models for the empirical and simulated datasets. The matrices represent the percentage of the 7200 datasets for which a pair of criteria in the corresponding column and row disagreed on. **a** represents the disagreement over the entire model (one of 24 models) while (**b–e**) represent disagreement over components of the models: **b** the substitution matrix that determines the substitution rates between the nucleotides, such that an equal parameter for all pairs defines JC and F81, two rates for transitions and transversions define K2P and HKY, and an independent parameter for each of the six pairs defines SYM and GTR, **c** the inclusion of the F component, i.e., equal base frequencies represent JC, K2P, and SYM, whereas unequal frequencies represent F81, HKY, and GTR, **d** the inclusion of the I parameter (proportion of invariable sites), **e** the inclusion of the G parameter (heterogeneous rates across sites following the gamma distribution). The percentages below and above the left diagonal represent the percentage of dissimilarities over empirical set and simulation set $c_0$, respectively. The percentages in the row of the BF criterion are among a subset of 1500 empirical datasets for which BF was computed (marked with an asterisk; see Methods). The analyses over this subset of 1500 datasets for all pairs of criteria is presented in Supplementary Figure 4

observed between AIC and AICc and even more so between BIC and DT. All criteria tended to agree more on the inclusion of the G parameter while the least agreement was on the chosen substitution matrix (Fig. 3b, e), partly due to the ternary selection compared to the binary selection posed by the other components (i.e., whether there is an equal rate parameter for all pairs of nucleotides, two independent rates for transitions and transversions, or an independent rate for each of the six pairs of nucleotides). Similar patterns were obtained for datasets that were simulated based on the sample sizes and estimated diversity rates of these empirical cases (Fig. 3a, upper triangle). Notably, there were fewer disagreements between the model selection criteria on the simulated data compared to the empirical data (Fig. 3a). We thus conclude that although the criteria may result in distinct selections, all best-fitted models lead to similar phylogenetic inferences.

**A fixed model instead of model selection.** Having established that alternative criteria for model selection have negligible effect on phylogenetic tree inference and ancestral sequence reconstruction, we next examined whether performing a model selection step prior to these inference procedures is essential. To this end, we reconstructed the phylogenies for all datasets using a fixed model. Specifically, we employed GTR+I+G and JC, representing the most complex and simplest models. We also compared performance when using the model that was used to simulate the data (i.e., the true model). While our initial hypothesis was that a model selection step would prove beneficial, our results pointed to the contrary. The percentages of correctly inferred tree topologies for the GTR+I+G model were highly similar to the model selection criteria across all simulation sets. Particularly, for simulation sets $c_0–c_2$, the percentages were highest for the GTR+I+G model and even better than under the true model (Table 2). Corroborating this observation, a similar trend emerged when the RF distances were examined (Table 3). Peculiarly, inference with the oversimplified model JC resulted in a reduction of only ~2% in the number of correctly inferred tree topologies compared to the other strategies, although analysis of the RF distances demonstrated that this decreased performance

was statistically significant (Tables 2 and 3; $p$-value < 0.05 when comparing JC to all other strategies across all simulation sets; pairwise Wilcoxon tests following the Bonferroni correction). This suggests that the most appropriate model is not of major importance for topology reconstruction, yet, the introduction of additional parameters may be beneficial. For branch-length inference, the average rankings of all model selection criteria, as well as the true model, were better than GTR+I+G across all examined simulation sets (Table 4; $p$-value < 0.05 when comparing GTR+I+G to all other strategies in simulations sets $c_0–c_2$; pairwise Wilcoxon with the Bonferroni correction), although the actual branch-length distances were highly similar (Supplementary Figure 3). The performance of the JC model was inferior to all other strategies in simulation sets $c_0–c_1$ and the corresponding branch-length distances were markedly larger compared to all other strategies (Supplementary Figure 3). We also examined the inferred distances for subsets of the data, binned according to tree size. In contrast to the negligible differences between the distances produced by the criteria, the true model, and GTR+I+G, the reconstruction with JC yielded large distances. This trend was preserved across increasing tree sizes, both for topological distances and branch-length distances (Supplementary Figures 2–3; Supplementary Data 2–3). The results obtained for the ancestral sequence reconstruction analysis using GTR+G (see Methods) were similar to those obtained using the established model selection criteria (Fig. 2 and Supplementary Table 1).

**The strategies performance across increasing alignment size.** One can speculate that the superiority of GTR+I+G for topology inference may not hold when small datasets are analyzed because of the possible incorporation of more error within each estimated parameter. To test this hypothesis, we compared the different strategies in subranges of the data, categorized by the alignment size. To this end, we binned the various datasets according to the number of taxa and the alignment length. The RF distances of GTR+I+G were similar to those of the model selection criteria across different data sizes (Fig. 4). In conclusion, our results suggest that at least for the tasks of phylogeny inference and ancestral sequence reconstruction, there is no clear benefit for

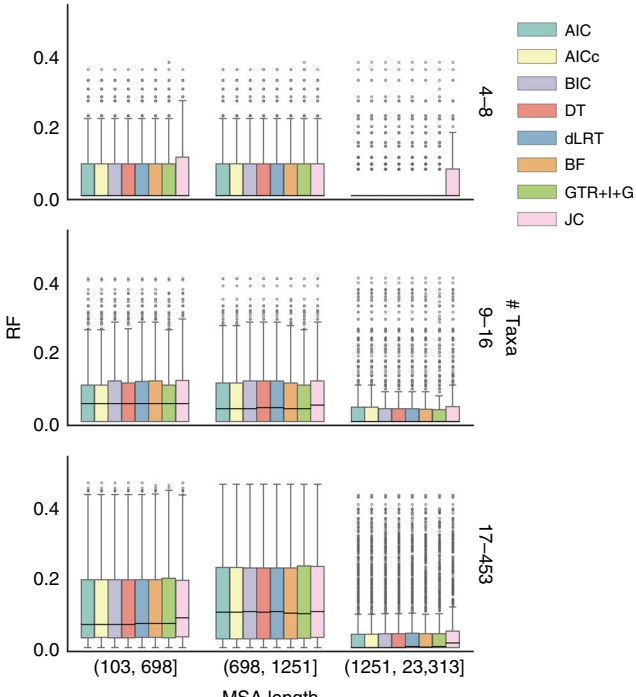

**Fig. 4** RF distances across different alignment sizes for simulation set $c_0$. RF distances ($y$ axes) were measured between trees reconstructed according to every strategy (denoted by the different colors) and the corresponding true trees. The data are binned according to the number of taxa (right-vertical axis) and alignment length (horizontal axis). The RF distances are divided by the number of nodes in the trees for a valid comparison across different tree sizes within a bin. The central horizontal lines represent the median values. The bounds of the boxes represent the first and third quartiles ($q1$ and $q3$, respectively). The whiskers extend to $(q3−q1) \times 1.5$ beyond the quartiles

performing a model selection step, and a fixed parameter-rich model can be used instead.

## Discussion

Model selection is considered as a fundamental step in the process of phylogeny reconstruction and has penetrated into the broad phylogenetic community, as implied by the ubiquitous use of the existing tools for model selection. Yet, as exemplified by our literature survey, there is much uncertainty surrounding the criterion of choice. Our initial goal in this study was to perform a rigorous analysis that could guide the community toward the preferred criteria in subranges of data characteristics. To this end, we examined the impact of various model selection criteria on phylogenetic inference over thousands of datasets that represent a range of realistic biological conditions. We evaluated the effect of using the various criteria on the reconstruction of the phylogenetic trees and on the inference of ancestral sequences by analyzing simulations of standard models and complex models that mimic empirical datasets more realistically. To our surprise, the results showed that all model selection criteria performed equivalently. Moreover, we found that using the most parameter-rich model, GTR+I+G, for all datasets instead of conducting a model selection step leads to phylogenies and ancestral sequences as accurate as those obtained when model selection is performed, suggesting that the standard practice of model selection prior to phylogeny inference is unnecessary when the currently used strategies are employed.

This study is not the first to investigate the importance of model selection and the relative effectiveness of the various

criteria[9,13,24–30]. Yet, previous studies have arrived at conflicting conclusions regarding the preferred criterion. A possible cause for this ambiguity could be the relatively restricted conditions that were used to simulate the data. Namely, several delimited values of sample size and sequence divergence were defined, and simulated data were generated according to combinations of them. Although this approach indeed varies the examined test cases and allows fair comparisons over different scenarios, it still does not reflect the complex patterns concealed within empirical datasets, particularly with the ongoing accumulation of data. To overcome this limitation, here we used realistic features that were extracted from thousands of empirical datasets, spanning three databases that differ in their biological context. The datasets in PlantDB[35] are typically of low divergence, as each contains homologous sequences within a single plant genus. The datasets in Selectome[36] contain homologous coding sequences across greater taxonomic groups, and were assembled for the inference of positive selection. These datasets thus exhibit rather high divergence but low number of taxa. PANDIT[37], on the other hand, covers common protein domains and thus the alignments are short but include many homologous protein-coding genes. Altogether, these form a diverse set of biological datasets that extends over a wide range of intricate realistic properties. Still, generating simulated datasets according to the properties of empirical datasets while relying on homogeneous substitution models does not reach the complexity of empirical datasets. To increase the complexity of these datasets, we generated additional simulation sets, $c_1$ and $c_2$, that integrate heterogeneity of the substitution models and evolutionary rates across sequence sites. An additional simulation set, $c_3$, was generated according to a codon model that illustrates processes which are dissimilar to those portrayed by the models available for inference. Our analyses over these assortments did not yield prominent differences between the criteria. Notably, even though these settings led to higher incongruencies, they still did not reach the intricacy of empirical datasets (Fig. 1), indicating that real data consist of patterns that are substantially more complex than the simple models commonly used for phylogeny reconstruction.

For some phylogenetic applications, the use of alternative models may not have much influence on the results, whereas for others, the selection of a best-fitted model might be beneficial. Previous studies[25–28,31] and the analyses conducted here revealed little impact of using alternative models on the accuracy of tree topologies. While our results were demonstrated for phylogenetic reconstruction and ancestral sequence reconstruction, evidence for the robustness of inference to the model employed was also shown for the estimation of relative evolutionary rates across proteins alignment sites[41], and for the inference of the evolutionary relationships when quartets are concerned[32]. We speculate that our conclusion should also hold for other tasks such as finding orthologous sequences, detecting horizontal gene transfer events, and the detection of conserved regions. For the inference of branch lengths, our results show that model selection leads to marginally more accurate estimates compared to a consistent use of GTR+I+G, and among the model selection criteria, the BIC and dLRT were most accurate. This finding complies with previous studies that sustained that using simpler models yields more accurate branch-length estimates[13,28,42]. This suggests that for divergence time estimation, choosing the best fit model using model selection criteria could contribute and should be further examined. Notably, none of the models can capture the true evolutionary processes, nor can they reconstruct precisely the true phylogeny[43–47]. Yet, in order to obtain more correct resolutions, considering additional models that parameterize other plausible processes could be beneficial. It should be noted that in this work only a specific set of commonly used nucleotide substitution

models were studied while the effect of other nucleotide substitution models as well as choosing among amino acid matrices and different codon models remains to be studied. Importantly, in some applications the benefit of using model selection is evident, e.g., when transition–transversion and GC-content biases are of interest[7] or for the inference of positive selection[48]. The main difference between the mentioned inferences, i.e., those that are robust to model selection versus those that might not be, is that in the latter the model selection is inherently important for the inference task, while in the former the substitution model can be regarded as a nuisance parameter.

While different model selection criteria differ in their chosen model, they select features of models that the data seem to support (Fig. 3). It is reasonable that the most parameter-rich model, which combines all of these components, would lead to similar inferences in the risk of including more noise. This raises the question whether any model could suffice. It has been previously shown that using an oversimplified model when the assumed evolutionary patterns are known to be violated deteriorates the accuracy of inference, and in such cases, complex models should be used[49–52]. Surprisingly, in our analysis the recovery rate of the true tree topology by JC was only ~2% lower than the rates obtained with the various model selection criteria, and this gap decreased for the more complex simulation sets (Table 2). The marginally inferior performance of JC is not specifically attributed to small or large trees, but is quite constant across all tree sizes (Supplementary Figure 2, Supplementary Data 2). These findings suggest that in many cases there are no major differences among the alternative models, and that any model can serve just as well. It has been previously shown that when topological uncertainties exist, reconstruction with the true model can result in an inaccurate topology while the reconstruction with a wrong model results in the accurate one[52,53]. In addition, our results suggest that the best-fitted models do not consistently yield topologies that are more accurate than using a fixed model (Tables 2 and 3). Namely, if model misspecification introduces a bias, it is not directional, and thus slightly reducing or increasing the level of misspecification does not lead to direct improvement in the accuracy of tree topology inference. Hence, choosing one model for phylogeny reconstruction performs quite similarly to others. Certainly, more theoretical research is needed to better understand the effect of alternative models on phylogeny reconstruction.

Admittedly, it is necessary to examine the confidence of using a certain model as a proof for the utility or irrelevance of model selection for various phylogenetic applications. A possible procedure could be to compare measures of model adequacy[54–61] or bootstrap support[62–64] across different model selection criteria. Ripplinger et al. have examined the absolute adequacy of the selected models, and found that they are supported in most cases[65]. Yet, these authors also found that even very simple models are not rejected and showed that the simplest models that were not rejected produced trees that are not significantly different from those produced using the best supported models. In spite of this, they claimed that model selection may become paramount when there are possible uncertainties in the topologies (i.e., Felsenstein or inverse-Felsenstein zones[49,51,66]). However, these conclusions were drawn from analyses over a small sample of 25 empirical datasets and 20 simulated datasets generated from only two sets of rate parameters. In order to obtain comprehensive conclusions, similar analyses should be conducted over a varied database such as the one used here. Due to the intensive computational work entailed with these procedures and the possible lack of power of these methods, we leave this to future work.

To conclude, our results imply that model selection may be unnecessary when one is interested in inferring ancestral sequences or in revealing the cladistic relationships among genes and organisms.

## Methods

**Empirical data assembly.** For a preliminary assessment of the discrepancies between the different model selection criteria, we assembled a database encompassing 7200 multiple sequence alignments (MSAs), 2400 from each of the following three databases: PlantDB[35], Selectome[36], and PANDIT[37]. The datasets in PlantDB were generated as described in Glick et al.[35], such that each MSA contains sequences belonging to a single plant genus and a potential outgroup. These MSAs contain between 2 and 912 species and span over 115–9417 aligned sites (Supplementary Figure 5a). The Selectome database[36] includes codon alignments of species within four groups (Euteleostomi, Primates, Glires, and Drosophila), which extend from 6 to 257 sequences and 72 to 64,734 aligned sites. The PANDIT database[37] includes alignments of protein sequences that extend from 2 to 2453 sequences and 15 to 6895 aligned sites. A subset of 2400 datasets was randomly selected from each database, excluding alignments that contained fewer than four sequences, fewer than 100 alignment sites, or produced low total divergence (i.e., when the multiplication of the total branch length by the alignment length is lower than 10) (Supplementary Figure 5b).

**Nucleotide substitution models.** The nucleotide substitution models examined in this study are the default 24 models assessed in jModelTest, i.e., JC[1], F81[2], K2P[3], HKY[4], SYM[5], and GTR[6] combined with the proportion of invariable sites (+I)[67,68], rate heterogeneity across sites (+G)[69], or both (+I+G). These models consist of several sets of parameters: the substitution matrix, the F, I, and G components. The substitution matrix describes the rates of substitution between every pair of nucleotides: JC and F81 assume an equal rate for all parameters; K2P and HKY assume two rates to distinguish between transitions and transversions; SYM and GTR assume an independent rate for each of the six pairs of substitutions. The F component describes whether the stationary nucleotide frequencies equal 0.25 (JC, K2P, and SYM), or are allowed to vary (F81, HKY, and GTR). The I parameter assesses the proportion of invariable sites, and the G parameter allows more flexibility by assigning heterogeneous rates across the alignment sites, drawn from a discrete gamma distribution with mean 1. The shape of the gamma distribution is a free parameter estimated from the data. For the application of ancestral sequence reconstruction, the combination of the I parameter was not examined since it is not implemented in PAML[70].

**Simulation set $c_0$: common models.** To analyze the performance of the six examined model selection criteria (Table 1), we conducted extensive simulations that characterized various evolutionary processes. To extend over a wide range of realistic data conditions, the input parameters were derived from the sampled empirical datasets. For each dataset, one model (termed hereafter the true model), was randomly selected out of a set of the 24 nucleotide substitution models. Given a single MSA, the phylogeny and the parameters required for generation of simulated data (alignments length, number of sequences, base frequencies, substitution rates, heterogeneity-across-sites, and proportion of invariant sites) were computed using PhyML[71]. Finally, INDELible[72] was executed over these parameters with the respective PhyML tree as the base tree, resulting in ca. 300 simulated alignments per model. In total, 7200 simulated datasets were generated (termed hereafter simulation set $c_0$).

**Simulation sets $c_1$ and $c_2$: integrate model misspecification.** To assess the performance of the model selection criteria over simulated data that incorporate more realistic patterns than the basic substitution models available for inference, we generated additional simulated datasets that integrated additional layers of complexity. The alignments for these analyses were derived from the 7200 empirical datasets used for the abovementioned simulations. We note that BF was not examined in these analyses due to intensive computations this procedure entails, and since it did not perform better than the ML criteria over simulation set $c_0$.

Simulation set $c_1$: Across-site variation of the substitution model. We generated 7200 simulated MSAs with heterogeneous substitution matrices across sites. To this end, each empirical dataset was divided to partitions of 50 sites (datasets were trimmed such that the alignment length is divisible by 50). Alignments were simulated using INDELible[72], such that the model and parameters used for each partition were derived from the respective partitions of the empirical data. Namely, jModelTest[33,34] was executed for each partition to obtain the best-fitted model and its inferred free parameters. To obtain the best-fitted model while avoiding a bias toward a particular criterion, one ML criterion was randomly selected per partition. A fixed BioNJ tree[73] reconstructed with the JC model (as implemented in PhyML) was used as the true tree for these simulations.

Simulation set $c_2$: Combination of model variation and rate heterogeneity across alignment sites. To increase the data complexity, we generated 7200 MSAs similar to $c_1$, with the addition of site-specific rates, drawn from distributions that are more complex than those assumed by current models (i.e., the I and G components). To this end, Rate4site[74] was executed for each empirical dataset to infer the evolutionary rate per position in the alignment using a Bayesian inference under the JC model. We used a fixed BioNJ tree[73], reconstructed with the JC model (as implemented in PhyML) as the true tree for these simulations. In order to combine distinct rates across alignment sites, we simulated multiple instances for every dataset as described for $c_1$, but each instance differed by the size of the tree, i.e., the

branch lengths, which were multiplied by one of the evaluated rates. Finally, the relevant sites were concatenated corresponding to every partition and rate.

**Simulation set $c_3$: a codon substitution model**. We generated 1000 simulated MSAs using a codon substitution model. To this end, we sampled 500 datasets from each database of coding sequences alignments, i.e., Selectome and PANDIT. For each dataset, a BioNJ tree[73] reconstructed with the JC model (as implemented in PhyML[71]) was used as an initial tree. CodeML application in the PAML package[70] was executed to optimize the branch lengths and the parameters that correspond to the M8 codon model[40]. These parameters were estimated as in Yang et al.[40]: the transition–transversion ratio parameter, the codon frequencies assuming the F3x4 codon frequency model, 11 site classes for the nonsynonymous–synonymous rate ratio such that 10 are drawn from the beta distribution and one additional class for positive selection. The optimized tree and inferred parameters were then used to simulate an alignment in the Evolver simulator in PAML[70]. The following model selection, tree reconstruction, and further analysis for these simulated datasets were performed using the nucleotide substitution models as was done for simulation sets $c_0$–$c_2$.

**Model selection and tree reconstruction**. jModelTest[33,34] was executed for each MSA to obtain the selections of AIC, AICc, BIC, and DT, with ML optimization of all parameters including branch lengths and tree topology. For the computation of the hierarchical likelihood ratio tests, we employed the dLRT criterion as implemented in jModelTest. Since dLRT assumes nested models along the decision process, the selections of dLRT were obtained by executing jModelTest with a fixed BioNJ tree[73]. To obtain the marginal likelihood estimates for BF calculation, the stepping-stone[19] algorithm implemented in RevBayes[75,76] was executed for each dataset and for each of the 24 models independently. The prior probabilities were determined according to the recommendations in RevBayes tutorials, as follows: parameters of the stationary base frequencies for F81, HKY, and GTR were assigned with equal probabilities; the prior probability for the transition–transversion ratio (kappa) parameter for K2P and HKY was specified from the lognormal distribution (mean = 0, std = 1.25); the substitution parameters for SYM and GTR were assigned with equal probabilities; the prior probability for the proportion of invariable sites (+I) was specified from the Beta distribution (with shape $\alpha = \beta = 0$); a diffuse prior for the alpha shape parameter of the gamma distribution for assessing the among-site-rate-variation (+G) was specified from the lognormal distribution (mean = 2, std = 0.587405; so that 95% of the prior density spans exactly one order of magnitude). The stepping-stone algorithm was executed using 50 categories of power posteriors, 10,000 generation of burn-in, and 1000 generations of running as was applied by Fan et al.[20]. Finally, the best model was selected as the one with maximal marginal likelihood. This was done for all datasets of $c_0$ and a subset of 1500 empirical datasets (500 from each database, due to long running times).

The trees were reconstructed for all the relevant models using PhyML[71] by ML optimization of the parameters, branch lengths, and topology (termed the selected trees). The reconstruction was accomplished according to nine strategies: six with the models selected by each of the criteria (i.e., AIC, AICc, BIC, DT, dLRT, and BF), one with the true model (i.e., the model that was used to simulate the data), and another two by consistently using the GTR+I+G model and the JC model.

**Tree comparison**. Tree comparison was performed by two metrics: topological and branch-length distances. Topologies were compared using the RF distance[38] as implemented in TreeCmp[77]. The branch lengths were compared using the BS distance[39] as implemented in Treedist[78]. First, for every pair of reconstruction strategies (see 'Model selection and tree reconstruction' section in Methods), we estimated their congruency by measuring the distances between the reconstructed trees across all datasets. Second, over simulated data, we quantified the discrepancy between the tree reconstructed with the selected model and the tree used for simulation (the selected tree and the true tree, respectively) by measuring the distances between them.

The RF and BS distances are relative to the size of the corresponding tree. Since the analyzed datasets consisted of varied data sizes and sequence divergences, the distances are not comparable across different trees. To enable comparison of the strategies performances across different trees, the strategies were ranked for each dataset from low distance (rank 1) to high distance. For simulation set $c_0$, nine strategies were evaluated including the five ML criteria, BF, reconstruction with the simplest and most complex models, i.e., JC and GTR+I+G, or with the true model that was used to simulate the data. For the complex simulations sets $c_1$–$c_2$, BF and the true model were not evaluated and thus the distances were ranked from 1 to 7. In case that several strategies obtained equal distances, they were assigned a rank that is the average of the ranks of those values. Note that the analysis of branch lengths was not performed for simulation set $c_3$ since the branch-length estimates are not comparable between the true and reconstructed trees, i.e., in the latter these represent substitutions per nucleotide site and in the former they represent substitutions per codon site.

**Ancestral sequence reconstruction**. In order to assess the impact of using alternative best-fitted models on the inference of ancestral sequences, we evaluated the pairwise distances between the simulated and reconstructed sequences at the root when using the best-fitted model and the reconstructed ML tree according to each model selection criterion. To root the input tree correctly, an outgroup is required. Since this information exists only for the PlantDB database, 1000 such datasets were sampled from the PlantDB database. The root was determined as the last common ancestor of the ingroup. To examine the effect of sequence divergence on the inference accuracy, we scaled the analyzed trees to a pre-defined set of total branch lengths (TBL). The values were defined according to the range of TBL values present in the 7200 analyzed datasets. Specifically, the following values were used: 0.08, 0.16, 0.27, 0.53, 1.19, 2.18, 3.5, 5.18, 9.5, which represent the deciles of the TBL in these datasets. For the task of ancestral sequence reconstruction, we opted to use an application which enables the inference with as many of the substitution models examined in our study. The six substitution matrices, i.e., JC, F81, K2P, HKY, SYM, and GTR are implemented in BaseML application in the PAML package[70], with or without the G parameter (heterogeneous rates across sites following the gamma distribution). Since the I parameter (proportion of invariant sites) is not implemented in PAML, only these 12 models were used for simulation and inference of the ancestral sequence. BaseML was run for each simulated dataset with the model selected by each criterion and the selected tree.

**Code availability**. The code for this study was written in python version 3.6. Computation of likelihood and parameter estimates, model selection, simulations, and tree comparison were executed using the following application versions: PhyML 3.0[71], RevBayes 1.0.6[75,76], PAML 4.8[70], jModelTest 2.1.7[33,34], Rate4site 3.2[74], INDELible 1.03[72], Treedist 1.0[78], and TreeCmp 1.0-b291[77]. The code for data simulation and inference has been deposited in Open Source Framework (OSF) with the identifier DOI 10.17605/OSF.IO/T3PF2[79].

**Reporting summary**. Further information on experimental design is available in the Nature Research Reporting Summary linked to this article.

## Data availability

The datasets contained within the empirical set and the four simulated sets ($c_0$–$c_3$) have been deposited in Open Source Framework (OSF) with the identifier DOI 10.17605/OSF.IO/T3PF2[79].

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

## Acknowledgements

S.A. is supported by Ph.D. fellowships provided by the Rothschild Caesarea Foundation and the Edmond J. Safra Center for Bioinformatics at Tel-Aviv University. D.A. is supported by a fellowship from the Fast & Direct Ph.D. Program by the Argentina Honors program. T.P. is supported by an Israeli Science Foundation number 802/16. I.M. is supported by an Israeli Science Foundation number 961/17.

## Author contributions

S.A. and D.A. jointly conceived the study, designed the work including simulations, inference, and analyses, and prepared the manuscript. I.M. and T.A. supervised this work and revised the manuscript for important conceptual advices.

## Additional information

**Competing interests:** The authors declare no competing interests.

