## [Peer Review File · Nature Communications]

Reviewers' Comments:

Reviewer #1:

Remarks to the Author:

Review by Jeff Thorne of "Is model selection a mandatory step for phylogeny reconstruction", a manuscript submitted by Abadi et al. to Nature Communications

Tens of thousands of published studies have employed statistical criteria to choose which probabilistic model of sequence evolution is most appropriate for analyzing particular data sets. Via both simulation and analysis of empirical data, the authors of this manuscript investigate whether the model selection criteria actually provide a benefit. The rather surprising conclusion is that these criteria do not have much practical value. In fact, the authors make the good point that large investments of computational time are devoted to model selection and these investments could be avoided by eliminating the model selection step.

As discussed in the manuscript, there are important caveats to this investigation of model selection criteria. Most importantly, this manuscript only considers model selection in the context of a relatively small number of simple probabilistic nucleotide substitution models. This focus is reasonable because the authors are concentrating on the nucleotide substitution models that are employed in the vast majority of evolutionary analyses. However, it is widely recognized that the widely-used models of sequence change are overly simple. Model selection may provide value for situations where more realistic (and more parameter-rich) models are investigated. To their credit, the authors raise this point.

Overall, I like this manuscript. It has several positive features. First, it addresses an issue that will be of substantial interest to the evolutionary biology community. Second, the work that is described is extensive. Many simulations are performed and the number of empirical data sets that were analyzed is especially impressive. Third, the interpretation of the results is sensible and is generally cautious.

I guess that model selection criteria might not all select the same model but that reasonable criteria might select features of models that the data seem to support. Therefore, I am not too surprised or bothered by the observation that different model selection criteria often choose select different models. I am also not too surprised or bothered by the observation that these different model selection criteria do not vary much in terms of how accurate is the resulting phylogeny inference. I find it reasonable that the GTR+I+G model performs about as well at phylogeny inference as do the model selection criteria. I was surprised by the fact that the very simple Jukes-Cantor model is only somewhat inferior in phylogeny inference to these other methods. I expected the JC model to be far worse. Is the "not terrible" performance of the Jukes-Cantor model attributable to the empirical data sets (and therefore also the simulated data sets) mostly not representing much sequence change? I do find it comforting that the "branch score" distance suffers as expected when the Jukes-Cantor model is assumed.

Even though this is a thorough and impressive study, I personally continue to believe the model selection criteria are worth the effort. As the authors note, not all evolutionary analyses have the main goal being phylogeny inference. Even for those studies where topology estimation is the main goal, I would need a demonstration that model selection criteria do not help with assessments of uncertainty.

In other words, maybe different substitution models will yield the same or similar inferred topologies but maybe measures of topological uncertainty will be misleading with some models (or model selection criteria) relative to others.

Likewise, maybe using model selection criteria improves the assessments of branch length uncertainty. Similarly, even if nearly identical ancestral sequences are inferred by different models, the uncertainties associated with these ancestral sequence inferences may vary and one model may lead to misleading uncertainty estimates. In this manuscript, I do not think it is reasonable to ask these authors to carefully assess the relationship between model selection criteria and uncertainty assessment. But, I do think they can devote more discussion to the issue.

While the manuscript is mostly easy to follow, one weakness is that the table legends and figure captions could be improved so that they are easier to follow without switching back-and-forth between the main text and the legends/tables.

The remainder of points that occur to me are mostly minor and I list them in the order in which they arise in the manuscript ...

Bottom of Page 4: remove "that" from "and that the chi-square" ?

Top of Page 9: With regard to ancestral sequence inference, there is a sentence that begins "For all criteria, in 97% ..." I do not quite understand this sentence. Can it be rephrased? Also and more importantly, I am not convinced that having greater than 95% sequence identity between inferred and true sequence is a persuasive argument that model selection criteria are not too important for ancestral sequence inference. Just a few mistakes might make an inferred ancestral sequence have very different functional properties than the actual ancestral sequence. Also, an inferred nucleotide state will be measured as correct whether that state had 51% probability or 99% probability.

Page 17 (the "Simulations" section): This section notes that about 300 simulated data sets corresponded to each of the 24 investigated models. Did the authors look at how model selection criteria performed by also separately analyzing each of the 24 subsets of size approximately 300? I am guessing that model selection is particularly important when much sequence change has occurred (i.e., large tree lengths). I am also guessing that phylogeny inference and branch lengths are most likely to be difficult to infer when the truth is a (somewhat) complicated model. It may be that this difficult case did not arise too often in the simulation study and that is why model selection criteria made so little difference. Admittedly, the authors did also look at more complicated evolutionary models (c1, c2, and c3). Still, separately analyzing results for the 24 subsets might have some value.

Page 19 (middle): "branch lengths distances" could be "branch length distances"

Page 19: The last sentence of the "Tree Comparison" section reads "The Bs distance of each dataset was divided by the sum of the branch lengths of the respective true tree, this enabling comparison of trees with difference number of branches and sequence divergence." I worry that this normalization (division) may make the branch score measure less useful for looking at how model selection criteria affect branch length estimates because I expect that it will give less weight to the data sets that represent lots of sequence divergence and that I expect to be most challenging for inference. However, I do not have an obvious alternative suggestion to make.

Page 25 (Figure 1 legend): This legend can be greatly improved by adding detail so that readers will understand exactly what the numbers represent. Much of the detail is in the main text but that can be moved to the legend. I can probably assume that the Bayes factor calculations assign equal prior probabilities to all models that were considered. But, this should be explicitly stated. Also, more detail about the Bayes factor procedure is needed. One could choose the tree topology with the highest

marginal probability (averaged over all substitution models). One could choose the substitution model with highest marginal probability (averaged over all topologies) and then find which topology is preferred by that substitution model. One could find the highest joint (marginal) probability of topology and substitution model. Finally, a striking feature of Figure 1b is that it shows that the actual empirical data sets yield far more disagreement among model selection criteria than any of the simple or complicated schemes. To me, this is yet more evidence that real data are not being modeled well by our simple models. Maybe some comments could be added about this in the main text of the manuscript.

Page 25 (Figure 3 legend): What about splitting the 1500 empirical BF results from the 7200 simulated BF results? Can comments be added about whether the results from the 1500 are similar to those from the 7200?

Page 29 (Figure 3): The fonts in parts b,c,d, and e are too small for old people to read.

Page 32 (Table 2): Replace "various models selection" with "various model selection" ? Also, some explanation in the Table legend should be made regarding to which simulations these numbers correspond.

Reviewer #2:

Remarks to the Author:

Please note that the attached review is formatted using Markdown.

Review of "Is model selection a mandatory step for phylogeny reconstruction?" by Abadi et al.

> By Sergei L Kosakovsky Pond (spond@temple.edu)

This is an important and timely paper which quite clearly demonstrates that some conventions in ubiquitous types of sequence analyses are effectively due to "urban legends". Considering that ModelTest is one of the Top 100 most cited papers of all time, is clearly used **all the time**. Yet, as shown by the authors here very convincingly, all we are really doing is (i) wasting energy and CPU cycles; (ii) creating a sense of false security in our inference. Indeed, just the literature survey part of this work, which report the lack of consensus and rigor in model selection (albeit statistical lackadaisical attitudes peruse our field, I am sad to say) is a very important contribution to the field. I strongly agree with the authors that model selection is largely unnecessary and confusing part of the textbook phylogeny inference approach, and hope that this paper will convince others to do the same. I liked the very thorough selection of representative datasets, logical series of experiments, and cleanly presented results. My specific comments and suggestions are shown below and can be considered **minor**.

Specific suggestions

* The one piece that is missing, I think, is at least **some discussion as to why models largely don't seem matter**.

* Introduction could be shortened a bit, I think.

* Perhaps cite additional lines of evidence for robustness of molecular evolutionary inference in the context of relative rate inference (<https://academic.oup.com/mbe/advance-article/doi/10.1093/molbev/msy127/5040133>) and branching resolution (<https://academic.oup.com/sysbio/advance-article-abstract/doi/10.1093/sysbio/syy047/5043533>)

> However, while richer models may capture the biological reality more accurately, they come with the risk of overfitting the examined observations [14]

There's actually very little evidence that overfitting (at least in the standard context of model selection) is a serious problem. The cited reference does not provide much support for this claim, other than general statements, and two references to "edge cases" with rarely used models (e.g. GTR + CAT_10 vs GTR + I + G).

> Notably, the ML criteria discussed above are aimed at obtaining the single best value of each parameter, and ignore any variations in their plausible values

I don't think this is relevant for model selection. Hypothesis testing and AIC incorporate estimation uncertainty by definition (otherwise larger Log L would result in accepting the model).

> Posada and Posada and Crandall initially concluded that methods that rely on likelihood ratio tests are more accurate than AIC and BIC.

I think these authors (especially David Posada) have moved on to advocating model averaging (in the non-Bayesian context), your ref 41, and the Bayesian crowd (e.g. Rambaut and Suchard) have long talked about models as components of the inference process, and their desire to "integrate the model out"

> In fact, each pair of criteria agreed on more than 92% of the 7,200 datasets (Fig 1a)

I don't see how Fig 1a supports this statement. It looks like you may be talking about **information** criteria only.

> ... because of the known inverse relationship between the number of free parameters and their standard errors

[?] I am only aware of the inverse relationship between the sample size and sampling variance; by increasing the number of parameters for a fixed sample size, you may or may not inflate the variance of said parameters; it really depends on the structure of the model and the relationship between model parameters.

> AIC-c and BIC

What is the sample size (n) for these criteria? That's not a trivial question because the literature is not very clear on it, and there is no good answer as far as I know. It is **not** simply the number of sites in the alignment (which is what you get for the tree reconstruction problem).

> ... JC led to mediocre performances as the correct tree topology was obtained in only 48.31% of the cases

Well, sure this is significant because of the large sample size, but you are talking about only a ~2%

drop in correct inference. I would call this **remarkable** that JC gets the answer right **nearly** as often as GTR+I+G.

> Importantly, in some applications the benefit of using model selection is evident, e.g., when comparing a model that allows for positive selection and a model that does not

I think a quote from George Box is appropriate here: "_Since all models are wrong the scientist must be alert to what is importantly wrong. It is inappropriate to be concerned about mice when there are tigers abroad_".

> These results suggest that for divergence time estimation, choosing the best fit model using these model selection criteria may be beneficial

Possibly, but in some cases if the model you **need** is not in the list of available models (e.g. a model that allows for variable selection), you will get low variance but high bias estimates (see <https://www.ncbi.nlm.nih.gov/pmc/articles/PMC3258043/>)

> PAML was then run for each simulated dataset with the models selected by each criterion and the selected tree...

PAML implements the marginal ancestral state reconstruction. I am a bit surprised you used it instead of the joint ML reconstruction, which was first described by Tal, as I recall. Perhaps you might discuss why you chose this method of AR.

> When you talk about AR of the **root** sequence. Considering that all models you discussed are reversible, how do you pick the root?

Simulations.

I think you need to conduct more simulations where the true model is **not** in the set of models you could select from. For example, you could simulate data under a codon model with some non-trivial selection profile and then infer trees using nucleotide models. Or use non-reversible models. Or use stem RNA (16x16) models. You get the idea... At the moment the simulations are still effectively combinations of nucleotide models.

Minor comments

>Over the last 50 years, a plethora of evolutionary models has been developed, each implying different hypotheses on the pattern of nucleotide evolution.

I don't think you want to say that models imply **hypotheses**, rather than represent different assumptions about how we think evolution operates and what is important to model.

> These, and other extensions such as accounting for the GC content, sum up to an excessive number of possible substitution schemes

The number is not **that** excessive; clearly we don't need that many models, but there are only 203 unique reversible matrices x small fixed number of frequency estimators x small fixed number of commonly used rate variation distributions.

> The increasing number of parameters grants the model the strength and flexibility to fit different

datasets and to capture their complexity

I don't think the adjective ****strong**** is something you can apply to a model.

> When the sample size is small compared to the number of parameters, it is advised to use the corrected version of AIC, termed AICc_{21,22}, since the former is only valid asymptotically as the size of the data approaches infinity

AIC-c is also only "asymptotically" valid (both AIC and AIC-c use a version of the central limit theorem for their derivation)/

> First, we observed that each of the six criteria selected models that managed to recover the topology of the true tree in 50-51% of the datasets (Table 2)

There's something not quite right with the grammar in this sentence.

> excluding alignments that contained less than four sequences, less than 100 alignment sites

****Fewer****, not ****less****, please.

> To allow for comparison across trees with different number of tips, the RF distance of each dataset was divided by the total number of splits

Why do you need to compare trees with different numbers of tips?

Reviewer #3:

Remarks to the Author:

In this manuscript, the authors examine the practice of model selection prior to phylogenetic inference. Their main questions concern comparisons of the various criteria used for model selection (e.g., AIC, AICc, BIC, etc.) and the impact of the criterion used on the accuracy of the inferred phylogeny. One of their main findings is that there is generally disagreement among criteria in selecting the model, but that the criteria all perform similarly in terms of the accuracy of the inferred phylogeny. They also consider the estimation of branch lengths and the inference of the ancestral (root) sequence. Again, differences in inference accuracy among the criteria were small, though in this case there were statistically significant differences among the models selected in terms of branch length accuracy. Finally, the authors compare use of a criterion for model selection to the specification of a complex model (i.e., GTR+I+G) without model selection, and find that the specification of GTR+I+G does not lead to a loss of accuracy in the inferred tree, though it may have an impact on the estimation of branch lengths. The authors thus recommend that the practice of model selection be abandoned in favor of a priori specification of a sufficiently complex model, saving the computational effort required in the model selection stage of the inference procedure.

The paper is well-written overall, and makes some important points, with which I agree in many cases. However, I think the significance and impact on phylogenetic practice may be overstated. For example, the authors state several times that they find it surprising that a consensus has not arisen in the literature with regard to which criterion should be used for model selection. But I think that a very plausible explanation for this is that empiricists are well-aware of the main finding of this paper — namely, that it's important to get approximately the correct model (e.g., we wouldn't want to use JC if the data arose from GTR+I+G, as the authors show), but once we get approximately the correct

model, the specific choice isn't very important, and thus the criterion used for model selection is not crucial. Second, it isn't really surprising (and definitely not "remarkable", as the authors state on line 263) that GTR+I+G works well in many cases. Since the parameters of the GTR+I+G model will be estimated during inference of the phylogeny and since the simpler models are nested within this one, if a simpler model was used to generate the data, we'd expect the parameter estimates to be similar to those under for the simpler model. So I agree completely that this is a reasonable thing to do, and this study bears this out.

I also have a few minor comments and wording issues, listed below by line number:

— line 29, "all criteria lead to similar inferences" — I think the authors need to be specific here. Upon reading the paper, we can see that this is certainly true for inference of the phylogeny, probably true for inference of the ancestral sequence, and possibly not true for inference of branch lengths. But there are other things we might also want to infer. For example, inference of changes in base frequency composition or transition/transversion bias may also be of interest. The authors later mention such possibilities.

— line 52, "excessive" might be too strong — might not evolution happen in a large number of different ways?

— In the paragraph that starts at line 63, I think that likelihood-based methods like AIC, BIC, etc., are referred to twice, at different places in the paragraph.

— line 91, the statement "reflects the statistical power of the comparison" is too vague; something more precise, for example, "the magnitude of the BF quantifies the relative strength of evidence for the two models", is needed.

— line 92, "since marginal likelihood is not a closed form expression ..." — but in general there may be a closed form for the marginal likelihood. This doesn't happen for problems in phylogenetics, but this statement makes it sound like that's generally true. Much of the remainder of this paragraph should be re-worded to be more precise, differentiating the phylogenetic setting from the basic statistical principles.

— line 141, "to sort" -> "to understand"

— line 141, remove comma after "criteria"

— line 256, remove the word "well"

— line 256, "evident" -> "evidenced"

— line 283, "branch lengths estimations" -> "branch length estimates"

— line 289, "nucleotide models" -> "nucleotide substitution models"

— line 301, "among genomes" is maybe too broad a statement — the paper doesn't really deal with genome-scale settings

— Table 1, remove the extra "for" in the description of the BF

Reviewer #1

Tens of thousands of published studies have employed statistical criteria to choose which probabilistic model of sequence evolution is most appropriate for analyzing particular data sets. Via both simulation and analysis of empirical data, the authors of this manuscript investigate whether the model selection criteria actually provide a benefit. The rather surprising conclusion is that these criteria do not have much practical value. In fact, the authors make the good point that large investments of computational time are devoted to model selection and these investments could be avoided by eliminating the model selection step.

As discussed in the manuscript, there are important caveats to this investigation of model selection criteria. Most importantly, this manuscript only considers model selection in the context of a relatively small number of simple probabilistic nucleotide substitution models. This focus is reasonable because the authors are concentrating on the nucleotide substitution models that are employed in the vast majority of evolutionary analyses. However, it is widely recognized that the widely-used models of sequence change are overly simple. Model selection may provide value for situations where more realistic (and more parameter-rich) models are investigated. To their credit, the authors raise this point.

Overall, I like this manuscript. It has several positive features. First, it addresses an issue that will be of substantial interest to the evolutionary biology community. Second, the work that is described is extensive. Many simulations are performed and the number of empirical data sets that were analyzed is especially impressive. Third, the interpretation of the results is sensible and is generally cautious.

We thank the reviewer for the supportive feedback and for the many productive insights. We accounted for all remarks in our revision and we address each one specifically as detailed below.

I guess that model selection criteria might not all select the same model but that reasonable criteria might select features of models that the data seem to support. Therefore, I am not too surprised or bothered by the observation that different model selection criteria often select different models. I am also not too surprised or bothered by the observation that these different model selection criteria do not vary much in terms of how accurate is the resulting phylogeny inference. I find it reasonable that the GTR+I+G model performs about as well at phylogeny inference as do the model selection criteria. I was surprised by the fact that the very simple Jukes-Cantor model is only somewhat inferior in phylogeny inference to these other methods. I expected the JC model to be far worse. Is the "not terrible" performance of the Jukes-Cantor model attributable to the empirical data sets (and therefore

also the simulated data sets) mostly not representing much sequence change? I do find it comforting that the "branch score" distance suffers as expected when the Jukes-Cantor model is assumed.

Following this comment, we now put more emphasis on the (surprisingly) moderate performance of JC.

We have revised the text in several sections. In the results (page 13) we write:

“Specifically, we employed GTR+I+G and JC, representing the most complex and simplest models. We also compared performance when using the model that was used to simulate the data (i.e., the true model). While our initial hypothesis was that a model selection step would prove beneficial, our results pointed to the contrary. The percentage of correctly inferred tree topologies was highest for the GTR+I+G model across all simulation sets, slightly better than that achieved for all other model selection criteria, and in particular better than under the true model (Table 2). Corroborating this observation, a similar trend emerged when the RF distances were examined, especially for simulation sets c_1 - c_3 (Table 3). Peculiarly, inference with the over-simplified model JC resulted in a reduction of only $\sim 2\%$ in the number of correctly inferred tree topologies compared to the other strategies, although analysis of the RF distances demonstrated that this decreased performance was statistically significant (Tables 2 and 3; p -value < 0.05 when comparing JC to all other strategies across all simulation sets; pairwise Wilcoxon tests following the Bonferroni correction). This suggests that the most appropriate model is not of major importance for topology reconstruction, yet, the introduction of additional parameters may be beneficial. For branch length inference, all model selection criteria, as well as the true model, performed better than a reconstruction with GTR+I+G or JC for c_0 (Table 4; p -value < 0.05 when comparing GTR+I+G and JC to all other strategies; pairwise Wilcoxon with the Bonferroni correction). Notably, the average ranking of GTR+I+G was 5.5 compared to ~ 4.7 of the other criteria, whereas that of JC was 6.46. Repeating these analyses over the complex simulation sets c_1 - c_2 resulted in very small differences between the strategies (Table 4). For c_3 , the superiority of criteria that tend toward more complex models (i.e., AIC and AICc) and especially that of GTR+I+G was more pronounced (Table 4, last column).”

The moderate performance of JC is not specifically attributed to datasets of certain size. To demonstrate this, we repeated the former analysis for subsets of the data, partitioned according to tree size (representing both the number of sequences and the amount of sequence divergence). Specifically, we binned the datasets according to the number of tree nodes when comparing RF distances, and according to the total branch lengths when comparing branch length distances. Then, we averaged the distances of all the datasets within each bin. This analysis showed that the RF distances are slightly larger for JC compared to the other strategies, and the size of this gap is preserved across all tree sizes (see Supplementary Table S2 and Supplementary Fig. S2). The analysis of branch estimates also showed that the distances of JC are larger than those of the other strategies, however this gap increased with the size of the tree (Supplementary Table S3 and Supplementary Fig. S3). We describe this analysis in the Results (following the quotation above, page 13):

“We also examined the inferred distances for subsets of the data, binned according to tree size. In contrast to the negligible differences between the distances produced by the criteria, the true

model, and GTR+I+G, the reconstruction with JC yielded large distances. This trend was preserved across increasing tree sizes, both for topological distances and branch length distances (Supplementary Figures S2-S3; Supplementary Tables S2-S3).”

In the Discussion (page 18) we write:

“It has been previously shown that using an oversimplified model when the assumed evolutionary patterns are known to be violated deteriorates the accuracy of inference, and in such cases, complex models should be used⁵⁸⁻⁶¹. Surprisingly, in our analysis the recovery rate of the true tree topology by JC was only ~2% lower than the rates obtained with the various model selection criteria, and this gap decreased for the more complex simulation sets (Table 2). The marginally inferior performance of JC is not specifically attributed to small or large trees, but is quite constant across all tree sizes (Supplementary Fig. S2 and Supplementary Table S2). These findings suggest that in many cases there are no major differences among the alternative models, and that any model can serve just as well.”

Even though this is a thorough and impressive study, I personally continue to believe the model selection criteria are worth the effort. As the authors note, not all evolutionary analyses have the main goal being phylogeny inference. Even for those studies where topology estimation is the main goal, I would need a demonstration that model selection criteria do not help with assessments of uncertainty. In other words, maybe different substitution models will yield the same or similar inferred topologies but maybe measures of topological uncertainty will be misleading with some models (or model selection criteria) relative to others.

Likewise, maybe using model selection criteria improves the assessments of branch length uncertainty. Similarly, even if nearly identical ancestral sequences are inferred by different models, the uncertainties associated with these ancestral sequence inferences may vary and one model may lead to misleading uncertainty estimates. In this manuscript, I do not think it is reasonable to ask these authors to carefully assess the relationship between model selection criteria and uncertainty assessment. But, I do think they can devote more discussion to the issue.

We agree. Following this comment, we added the following paragraph to the Discussion (pages 18-19):

“Admittedly, it is necessary to examine the confidence of using a certain model as a proof for the utility or irrelevance of model selection for various phylogenetic applications. A possible procedure could be to compare measures of model adequacy⁶³⁻⁷⁰ or bootstrap support⁷¹⁻⁷³ across different model selection criteria. Ripplinger et al.⁷⁴ have examined the absolute adequacy of the selected models, and found that they are supported in most cases⁷⁴. Yet, these authors also found that even very simple models are not rejected and showed that the simplest models that were not rejected produced trees that are not significantly different from those produced using the best supported models. In spite of this, they claimed that model selection may become

paramount when there are possible uncertainties in the topologies (i.e., Felsenstein^{58,60,75} or inverse-Felsenstein zones^{58,60}). However, these conclusions were drawn from analyses over a small sample of 25 empirical datasets and 20 simulated datasets generated from only two sets of rate parameters. In order to obtain comprehensive conclusions, similar analyses should be conducted over a varied database such as the one used here. Due to the intensive computational work entailed with these procedures and the possible lack of power of these methods, we leave this to future work.”

While the manuscript is mostly easy to follow, one weakness is that the table legends and figure captions could be improved so that they are easier to follow without switching back-and-forth between the main text and the legends/tables.

We thank the reviewer for this remark. We made an effort to improve the tractability of the manuscript, hence, we modified the figure legends, the methods, and the details given in the Results section. We believe the revised manuscript is easier to follow.

The remainder of points that occur to me are mostly minor and I list them in the order in which they arise in the manuscript ...

Bottom of Page 4: remove "that" from "and that the chi-square" ?

We shortened the introduction and omitted this sentence.

Top of Page 9: With regard to ancestral sequence inference, there is a sentence that begins "For all criteria, in 97% ..." I do not quite understand this sentence. Can it be rephrased? Also and more importantly, I am not convinced that having greater than 95% sequence identity between inferred and true sequence is a persuasive argument that model selection criteria are not too important for ancestral sequence inference. Just a few mistakes might make an inferred ancestral sequence have very different functional properties than the actual ancestral sequence. Also, an inferred nucleotide state will be measured as correct whether that state had 51% probability or 99% probability.

We believe this comment stems from our misleading phrasing. As pointed by the reviewer, the percentage of similarity cannot be used to determine if the inference is good or not. However, the aim of this analysis was to show that different criteria (and thus different best-fitted models) lead to similar results. We rephrased the referenced sentence and emphasized this objective. In addition, the figure labels and legend were modified accordingly. In page 9 we write:

“Subsequently, we examined whether the use of different model selection criteria has an effect on ancestral sequence reconstruction, as an example of an analysis which is downstream to phylogeny inference. To this end, each of the selected models (together with their corresponding selected trees) was used to infer the root sequence for 1,000 datasets (see Methods). Then, we measured the percentage of incorrectly inferred sites of each inferred sequence compared to the corresponding true sequence. The different criteria produced highly similar results. Namely, for all criteria, in 44% of the datasets the inferred sequence was identical to the true one, and in 97% of the datasets fewer than 5% of the sites were erroneous. Even though the sequence divergences in these simulated datasets reflect those found in many empirical datasets, we hypothesized that noticeable differences would become apparent when more divergent sequences, representing more challenging inference cases, were simulated. To this end, we resized all trees to several scales and repeated the analysis. For all criteria, the average percentage of incorrectly inferred sites increased with the increase in sequence divergence, however, the dissimilarities between every pair of the inferred sequences were still negligible (Fig. 2a and Supplementary Table S4). This suggests that choosing among model selection criteria has minor effect on the accuracy of ancestral sequence reconstruction.”

Page 17 (the "Simulations" section): This section notes that about 300 simulated data sets corresponded to each of the 24 investigated models. Did the authors look at how model selection criteria performed by also separately analyzing each of the 24 subsets of size approximately 300? I am guessing that model selection is particularly important when much sequence change has occurred (i.e., large tree lengths). I am also guessing that phylogeny inference and branch lengths are most likely to be difficult to infer when the truth is a (somewhat) complicated model. It may be that this difficult case did not arise too often in the simulation study and that is why model selection criteria made so little difference. Admittedly, the authors did also look at more complicated evolutionary models (c1, c2, and c3). Still, separately analyzing results for the 24 subsets might have some value.

We agree with all the points raised above. In the original submission, we presented the average distance across all datasets, after dividing them by the tree size. While this procedure enabled the comparison of distances for trees of different sizes (generated from datasets of different sizes), it also obscured the performances of the different strategies in specific ranges of the data and the magnitude of those distances as the reviewer implied. In order to resolve these potential problems, we revised the analyses in several manners. First, instead of presenting the average normalized distance across all datasets, we now present the average ranking of the strategies. This revised analysis thus reflects the average performance rather than the average distance. Second, in order to reflect the magnitude of the distances and the effect of more complex cases, we now present the average distances (without normalization) across different subsets, binned according to tree sizes (Supplementary Figures S2-S3 and Supplementary Tables S2-S3). This enables both to aggregate distances of datasets of similar tree sizes and to examine the effect

of tree size on the magnitude of those distances. Since the tree size reflects the data size as well as the complexity of the data, we find this method of binning more effective than separating to the 24 models.

Page 19 (middle): "branch lengths distances" could be "branch length distances"

Fixed.

Page 19: The last sentence of the "Tree Comparison" section reads "The Bs distance of each dataset was divided by the sum of the branch lengths of the respective true tree, thus enabling comparison of trees with difference number of branches and sequence divergence." I worry that this normalization (division) may make the branch score measure less useful for looking at how model selection criteria affect branch length estimates because I expect that it will give less weight to the data sets that represent lots of sequence divergence and that I expect to be most challenging for inference. However, I do not have an obvious alternative suggestion to make.

We agree that this normalization gave more emphasis to trees with short branch lengths and believe that the alternative analyses we present in this revised submission resolve this issue since we now present the results within each bin of tree size (as described in response to the previous comment). This modified presentation also enables elucidating the effect of tree size on the distances obtained by the different strategies. The distances across different tree sizes are demonstrated and provided in Supplementary Figures S2 and S3, and Supplementary Tables S2 and S3.

Page 25 (Figure 1 legend): This legend can be greatly improved by adding detail so that readers will understand exactly what the numbers represent. Much of the detail is in the main text but that can be moved to the legend.

The legend of figure 1 was expanded and more details were added to make it stand on its own:

“Figure 1. Pairwise distances between the trees inferred by the evaluated strategies. The number within each cell represents the percentage of discrepancies between the two strategies at the row and column. The best fitted model was computed for each criterion, and the trees were reconstructed using ML optimizations according to this model, as well as for the most complex and simplest models – GTR+I+G and JC. For each pair of strategies (rows and columns) the percentage of identical trees over 7,200 datasets are presented (see * and ** below). The upper

right triangles represent the percentages of different topologies and the lower left triangles represent different branch length estimates. Clearly, two different models lead to different branch lengths estimates, hence the latter reflects the percentage of differently selected models. The matrices represent the following datasets: (a) simulation set c_0 , (b) the empirical set, (c) simulation set c_1 , (d) simulation set c_2 , and (e) simulation set c_3 . (*) The percentages in the row and column of the BF criterion in panel b were computed over a subset of 1,500 empirical datasets for which BF was computed (marked with an asterisk; see Methods). The analysis over this subset of 1,500 datasets for all comparisons is presented in Supplementary Fig. S1. (**) The percentages of the simulation set c_3 were computed over a subset of 1,000 datasets that represent coding sequences (see Methods)."

I can probably assume that the Bayes factor calculations assign equal prior probabilities to all models that were considered. But, this should be explicitly stated. Also, more detail about the Bayes factor procedure is needed. One could choose the tree topology with the highest marginal probability (averaged over all substitution models). One could choose the substitution model with highest marginal probability (averaged over all topologies) and then find which topology is preferred by that substitution model. One could find the highest joint (marginal) probability of topology and substitution model.

Fixed. We added more details in the Methods section to explain explicitly how BF was computed. In pages 23-24 we now write:

"To obtain the marginal likelihood estimates for BF calculation, the 'stepping stone'²⁶ algorithm implemented in RevBayes^{84,85} was executed for each dataset and for each of the 24 models independently. The prior probabilities were determined according to the recommendations in RevBayes tutorials, as follows: parameters of the stationary base frequencies for F81, HKY, and GTR were assigned with equal probabilities; the prior probability for the transition-transversion ratio (kappa) parameter for K2P and HKY was specified from the lognormal distribution, (mean=0, std=1.25); the substitution parameters for SYM and GTR were assigned with equal probabilities; the prior probability for the proportion of invariable sites (+I) was specified from the Beta distribution (with shape $\alpha=\beta=0$); a diffuse prior for the alpha shape parameter of the gamma distribution for assessing the among-site-rate-variation (+G) was specified from the lognormal distribution (mean=2, std 0.587405; so that 95% of the prior density spans exactly one order of magnitude). The stepping stone algorithm was executed using 50 categories of power posteriors, 10,000 generation of burn-in, and 1,000 generations of running as was applied by Fan et al.²⁷. Finally, the best model was selected as the one with maximal marginal likelihood. This was done for all datasets of c_0 and a subset of 1,500 empirical datasets (500 from each database, due to long running times)."

A striking feature of Figure 1b is that it shows that the actual empirical data sets yield far more disagreement among model selection criteria than any of the simple or complicated schemes. To me, this is yet more evidence that real data are not being modeled well by our simple models. Maybe some comments could be added about this in the main text of the manuscript.

Following this important comment, we now better emphasize that empirical datasets yield far more disagreement than the simulated ones. We first note this in the Results section as a motivation for simulations of more complex sets and elaborate more on this in the Discussion (pages 15-16):

“However, generating simulated datasets according to the properties of empirical datasets while relying on homogeneous substitution models still does not reach the complexity of empirical datasets. To increase the complexity of these datasets, we generated additional simulation sets, c_1 and c_2 , that integrate heterogeneity of the substitution models and evolutionary rates across sequence sites. An additional simulation set, c_3 , was generated according to a codon model that illustrates processes which are dissimilar to those portrayed by the models available for inference. Our analyses over these assortments did not yield prominent differences between the criteria. Notably, even though these settings led to higher incongruencies, they still did not reach the intricacy of empirical datasets (Fig. 1), indicating that real data consist of patterns that are substantially more complex than the simple models commonly used for phylogeny reconstruction.”

Page 25 (Figure 3 legend): What about splitting the 1500 empirical BF results from the 7200 simulated BF results? Can comments be added about whether the results from the 1500 are similar to those from the 7200?

Following this suggestion, we reproduced these figures for the filtered dataset, such that the percentages are computed for the subset of 1,500 datasets of all simulation sets. These figures are now provided in the supplementary information as Supplementary Fig. S1 and S4 (complementary to Fig. 1 and Fig. 3). These filtered matrices are very similar to the ones produced over the 7,200 datasets, suggesting that this sample suffices in order to draw similar conclusions. Additionally, we modified the legends of Fig. 1 and Fig. 3 to clarify the difference between the comparisons of BF and the other strategies.

Page 29 (Figure 3): The fonts in parts b,c,d, and e are too small for old people to read.

Fixed. Our apologies.

Page 32 (Table 2): Replace "various models selection" with "various model selection"? Also, some explanation in the Table legend should be made regarding to which simulations these numbers correspond.

Fixed. In the previous version, the results presented in Tables 2 and 3 were separated according to the simulation sets, and for each set the tables presented the summary of the statistics (%Correct topologies, RF distance, Bs distance). In the revised manuscript, we present these data in several tables, such that each table presents the results of a different statistic across all simulation sets: in table 2 we provide the percentages of correct topologies, in table 3 the average ranks of topological distances, and in table 4 the average ranks of branch length distances. Within each table, we provide the results obtained for the different simulation sets (c_0 - c_3).

Reviewer #2

This is an important and timely paper which quite clearly demonstrates that some conventions in ubiquitous types of sequence analyses are effectively due to "urban legends". Considering that ModelTest is one of the Top 100 most cited papers of all time, is clearly used ****all the time****. Yet, as shown by the authors here very convincingly, all we are really doing is (i) wasting energy and CPU cycles; (ii) creating a sense of false security in our inference. Indeed, just the literature survey part of this work, which report the lack of consensus and rigor in model selection (albeit statistical lackadaisical attitudes perfuse our field, I am sad to say) is a very important contribution to the field. I strongly agree with the authors that model selection is largely unnecessary and confusing part of the textbook phylogeny inference approach, and hope that this paper will convince others to do the same. I liked the very thorough selection of representative datasets, logical series of experiments, and cleanly presented results. My specific comments and suggestions are shown below and can be considered ****minor****.

We thank the reviewer for the positive feedback and for all the insightful comments. We accounted for every remark as detailed below and revised the manuscript accordingly.

Specific suggestions

*** The one piece that is missing, I think, is at least ****some discussion as to why models largely don't seem matter****.**

Done. We elaborate on this issue in the Discussion (page 18):

“While different model selection criteria differ in their chosen model, they select features of models that the data seem to support (Fig. 3). It is reasonable that the most parameter-rich model, which combines all of these components, would lead to similar inferences in the risk of including more noise. This raises the question whether any model could suffice. It has been previously shown that using an oversimplified model when the assumed evolutionary patterns are known to be violated deteriorates the accuracy of inference, and in such cases, complex models should be used^{58–61}. Surprisingly, in our analysis the recovery rate of the true tree topology by JC was only ~2% lower than the rates obtained with the various model selection criteria, and this gap decreased for the more complex simulation sets (Table 2). The marginally inferior performance of JC is not specifically attributed to small or large trees, but is quite constant across all tree sizes (Supplementary Fig. S2 and Supplementary Table S2). These findings suggest that in many cases there are no major differences among the alternative models, and that any model can serve just as well. Evidently, more theoretical research is required to better understand the effect of

alternative models on phylogeny reconstruction. It has been previously shown that when topological uncertainties exist, reconstruction with the true model can result in an inaccurate topology while the reconstruction with a wrong model results in the accurate one^{61,62}. In addition, our results suggest that the best fitted models do not consistently yield topologies that are more accurate than using a fixed model. Thus, if model misspecification introduces a bias, it is not directional and we cannot know a-priori whether this bias will decrease or increase the accuracy of estimation. Hence, overall, choosing a fixed model for reconstruction performs quite similarly to others.”

*** Introduction could be shortened a bit, I think.**

We made efforts to make the introduction more succinct while conserving the details that are important for following the manuscript. To this end, we removed the detailed description of the substitution models, and generalized the description of the criteria.

*** Perhaps cite additional lines of evidence for robustness of molecular evolutionary inference in the context of relative rate inference (<https://academic.oup.com/mbe/advance-article/doi/10.1093/molbev/msy127/5040133>) and branching resolution (<https://academic.oup.com/sysbio/advance-article-abstract/doi/10.1093/sysbio/syy047/5043533>)**

We thank the reviewer for pointing these studies. Indeed, these articles are supporting evidence for our findings, and we added them to the Discussion (page 16):

“For some phylogenetic applications, the use of alternative models may not have much influence on the results, whereas for others, the selection of a best-fitted model might be beneficial. Previous studies^{33–36,39} and the analyses conducted here revealed little impact of using alternative models on the accuracy of tree topologies. While our results were demonstrated for phylogenetic reconstruction and ancestral sequence reconstruction, evidence for the robustness of inference to the model employed was also shown for the estimation of relative evolutionary rates across proteins alignment sites⁵⁰, and for the inference of the evolutionary relationships when quartets are concerned⁴⁰. We speculate that our conclusion should also hold for other tasks such as finding orthologous sequences, detecting horizontal gene transfer events, and the detection of conserved regions.”

> **However, while richer models may capture the biological reality more accurately, they come with the risk of overfitting the examined observations [14]**

There's actually very little evidence that overfitting (at least in the standard context of model selection) is a serious problem. The cited reference does not provide much support for this claim, other than general statements, and two references to "edge cases" with rarely used models (e.g. GTR + CAT_10 vs GTR + I + G).

Fixed. This statement was rephrased as follows (page 3):

“Accounting for more parameters grants a model the flexibility to fit different datasets and capture their complexity. However, the expected error of each estimate increases with the increase in the number of parameters, which is problematic mainly when data are scarce.”

> **Notably, the ML criteria discussed above are aimed at obtaining the single best value of each parameter, and ignore any variations in their plausible values**

I don't think this is relevant for model selection. Hypothesis testing and AIC incorporate estimation uncertainty by definition (otherwise larger Log L would result in accepting the model).

Indeed, the information criteria do account for the estimation uncertainty, but this is done through penalizing the number of parameters and not the processes they represent. That is, two models that have the same amount of parameters, e.g., SYM and HKY+I are penalized similarly regardless of the types of parameters each of them incorporates or their plausible variances. This is accounted for under a Bayesian approach. We rephrased this sentence as follows (page 4):

“Notably, handling the uncertainty within model testing by the ML criteria depicted above is accomplished by accounting for the number of parameters assessed in the computation, but not for the type of processes they represent. For example, the penalty for a parameter that distinguishes between transition and transversion would be identical to the penalty imposed for a parameter that assesses the number of invariant sites. In contrast, under the Bayesian approach, model selection can be performed using the marginal likelihood, which is the probability of the data given the model, while marginalizing the estimates (Table 1).”

> Posada and Posada and Crandall initially concluded that methods that rely on likelihood ratio tests are more accurate than AIC and BIC.

I think these authors (especially David Posada) have moved on to advocating model averaging (in the non-Bayesian context), your ref 41, and the Bayesian crowd (e.g. Rambaut and Suchard) have long talked about models as components of the inference process, and their desire to "integrate the model out"

This is true, and a detail we neglected to mention. Our aim in this paragraph was to reflect the ambiguity in the preferred criterion, and in particular when the task is to uncover the generating model. Following this comment, we rephrased the paragraph to clarify these two points:

“Nevertheless, as far as accuracy of choosing the generating model is concerned, there appears no consensus regarding the preferred criterion. Posada³⁸ and Posada and Crandall³¹ initially concluded that methods that rely on likelihood ratio tests perform better than AIC and BIC. However, a later study by Posada and Buckley concluded that the use of hLRT may not be effective for real data and therefore averaging different models according to the weights given by AIC or BIC is preferred³⁷. Increasing this ambiguity, an additional study showed that BIC and DT select the generating model more frequently than AIC and hLRT³², whereas under other simulation conditions, AIC was shown to be more accurate than BIC³⁴. Notably, these studies did not thoroughly examine the various tasks that are downstream to model selection. Hence, it is unclear whether the use of alternative best-fitted models according to different criteria would result in different inferences. It was argued that the inferred topology should be quite robust to the selected model^{33–36,39}, yet other applications, such as branch lengths estimation and ancestral sequence reconstruction, may be more sensitive^{18,35,36,38,40}. “

> In fact, each pair of criteria agreed on more than 92% of the 7,200 datasets (Fig 1a)

I don't see how Fig 1a supports this statement. It looks like you may be talking about ****information**** criteria only.

Fixed. BF had the maximal incongruency of 17% with any other criterion. We corrected the indicated percentage to 83%.

> ... because of the known inverse relationship between the number of free parameters and their standard errors

[?] I am only aware of the inverse relationship between the sample size and sampling variance; by increasing the number of parameters for a fixed sample size, you may or may not inflate the variance of said parameters; it really depends on the structure of the model and the relationship between model parameters.

We agree. We rephrased this motivation more accurately:

“One can speculate that the superiority of GTR+I+G for topology inference may not hold when small datasets are analyzed because of the possible incorporation of more error within each estimated parameter.”

> AIC-c and BIC

What is the sample size (n) for these criteria? That's not a trivial question because the literature is not very clear on it, and there is no good answer as far as I know. It is **not**** simply the number of sites in the alignment (which is what you get for the tree reconstruction problem).**

Indeed, this issue forms one of the basic problems of employing model selection to phylogenetics. Notably, the data size is considered as the number of independent data particles included in the dataset. However, the sites as well as the sequences in an MSA are evidently dependent. The computations of those criteria in our study were done in jModelTest, in which the data size is determined by the number of sites in the alignment. We added this information to Table 1.

> ... JC led to mediocre performances as the correct tree topology was obtained in only 48.31% of the cases

Well, sure this is significant because of the large sample size, but you are talking about only a ~2% drop in correct inference. I would call this **remarkable**** that JC gets the answer right ****nearly as often as GTR+I+G****.**

Following this important comment we rephrased the text to emphasize that JC was only slightly inferior to the other strategies. In the Results, page 13 we write:

“Specifically, we employed GTR+I+G and JC, representing the most complex and simplest models. We also compared performance when using the model that was used to simulate the data (i.e., the true model). While our initial hypothesis was that a model selection step would prove beneficial, our results pointed to the contrary. The percentage of correctly inferred tree topologies was highest for the GTR+I+G model across all simulation sets, slightly better than that achieved for all other model selection criteria, and in particular better than under the true model (Table 2). Corroborating this observation, a similar trend emerged when the RF distances were examined, especially for simulation sets c_1 - c_3 (Table 3). Peculiarly, inference with the oversimplified model JC resulted in a reduction of only ~2% in the number of correctly inferred tree topologies compared to the other strategies, although analysis of the RF distances demonstrated that this decreased performance was statistically significant (Tables 2 and 3; *p-value* < 0.05 when comparing JC to all other strategies across all simulation sets; pairwise Wilcoxon tests following the Bonferroni correction). This suggests that the most appropriate model is not of major importance for topology reconstruction, yet, the introduction of additional parameters may be beneficial.”

We also discuss these results in page 17:

“While different model selection criteria differ in their chosen model, they select features of models that the data seem to support (Fig. 3). It is reasonable that the most parameter-rich model, which combines all of these components, would lead to similar inferences in the risk of including more noise. This raises the question whether any model could suffice. It has been previously shown that using an oversimplified model when the assumed evolutionary patterns are known to be violated deteriorates the accuracy of inference, and in such cases, complex models should be used⁵⁸⁻⁶¹. Surprisingly, in our analysis the recovery rate of the true tree topology by JC was only ~2% lower than the rates obtained with the various model selection criteria, and this gap decreased for the more complex simulation sets (Table 2). The marginally inferior performance of JC is not specifically attributed to small or large trees, but is quite constant across all tree sizes (Supplementary Fig. S2 and Supplementary Table S2). These findings suggest that in many cases there are no major differences among the alternative models, and that any model can serve just as well.”

> Importantly, in some applications the benefit of using model selection is evident, e.g., when comparing a model that allows for positive selection and a model that does not

I think a quote from George Box is appropriate here: “_Since all models are wrong the scientist must be alert to what is importantly wrong. It is inappropriate to be concerned about mice when there are tigers abroad_”.

We thank the reviewer for the suggestion. We refer to the source of this citation (reference number 28) in the Introduction:

“Obviously, no evolutionary model can fully capture the genuine complexity of the evolutionary process, such that even the most adequate one merely provides an approximation of reality²⁸.”

> **These results suggest that for divergence time estimation, choosing the best fit model using these model selection criteria may be beneficial**

Possibly, but in some cases if the model you *need*** is not in the list of available models (e.g. a model that allows for variable selection), you will get low variance but high bias estimates (see <https://www.ncbi.nlm.nih.gov/pmc/articles/PMC3258043/>)**

We agree that this sentence should be better stated. We used the suggested reference and expanded the discussion on this. In pages 16-17 we write:

“This inconsistency suggests that for divergence time estimation, choosing the best fit model using model selection criteria should be further examined. Notably, none of the models can capture the true evolutionary processes, nor can they reconstruct precisely the true phylogeny⁵²⁻⁵⁶. Yet, in order to obtain more correct resolutions, considering additional models that parameterize other plausible processes could be beneficial. It should be noted that in this work only a specific set of commonly used nucleotide substitution models were studied while the effect of other nucleotide substitution models as well as choosing among amino acid matrices and different codon models remains to be studied. Importantly, in some applications the benefit of using model selection is evident, e.g., when transition-transversion and GC-content biases are of interest⁷ or for the inference of positive selection⁵⁷. The main difference between the mentioned inferences, i.e., those that are robust to model selection versus those that might not be, is that in the latter the model selection is inherently important for the inference task, while in the former the substitution model can be regarded as a nuisance parameter.”

> **PAML was then run for each simulated dataset with the models selected by each criterion and the selected tree...**

PAML implements the marginal ancestral state reconstruction. I am a bit surprised you used it instead of the joint ML reconstruction, which was first described by Tal, as I recall. Perhaps you might discuss why you chose this method of AR.

After a thorough search, we chose to use PAML for ancestral sequence reconstruction since it implements most of the models examined in this study. Unlike other ASR applications, in which only several substitution models are implemented, PAML allows for the estimation of all the six substitution matrices, i.e., JC, F81, K2P, HKY, SYM, and GTR, with or without the +G component. We added some information to the Methods section. In page 26 we write:

“For the task of ancestral sequence reconstruction, we opted to use an application which enables the inference with as many of the substitution models examined in our study. The six substitution matrices, i.e., JC, F81, K2P, HKY, SYM, and GTR are implemented in BaseML application in the PAML package⁷⁹, with or without the ‘+G’ component. Since the ‘+I’ component (proportion of

invariant sites) is not implemented in PAML, only these 12 models were used for simulation and inference of the ancestral sequence. BaseML was run for each simulated dataset with the model selected by each criterion and the selected tree.”

> When you talk about AR of the **root**** sequence. Considering that all models you discussed are reversible, how do you pick the root?**

The datasets that were used for the ancestral sequence reconstruction analysis were sampled only among those from PlantDB which have an outgroup of a single taxon that was used to root the tree. The root was then determined as the root of the ingroup. We added this to the Methods section. In page 25 we write:

“To root the input tree correctly, an outgroup is required. Since this information exists only for the PlantDB database, 1,000 such datasets were sampled from the PlantDB database. The root was determined as the last common ancestor of the ingroup.”

Simulations.

I think you need to conduct more simulations where the true model is **not**** in the set of models you could select from. For example, you could simulate data under a codon model with some non-trivial selection profile and then infer trees using nucleotide models. Or use non-reversible models. Or use stem RNA (16x16) models. You get the idea... At the moment the simulations are still effectively combinations of nucleotide models.**

Following this suggestion, we performed another set of simulations, generated according to a codon site model, M8. The inferences and analyses were performed similar to the other simulation sets, using nucleotide substitution models. Evidently, the patterns of the inferred results were similar to the ones obtained for the other simulation sets, with the exception that here GTR+I+G appeared as the best model also for branch length estimates. These results were added to Tables 2-4, and are summarized in pages 10-11:

“In spite of the enhanced complexity, the simulation sets presented above were still generated based on homogeneous nucleotide substitution models used for inference. In order to examine whether analyses over data that were generated based on other evolutionary patterns are in line with the deduction above, we used a codon model, M8^{48,49}, to simulate an additional set termed c_3 . The rates that were used to simulate these datasets were inferred from a subset of 1,000 alignments of coding genes included in the empirical set. The succeeding analyses over these generated codon alignments were performed using the nucleotide substitution models, similar to c_0 - c_2 . As before, the percentages of accurate topologies obtained by all criteria were highly similar (Table 2, last column). It should be noted that these percentages were higher than those of

simulation sets c_0-c_2 , perhaps due to the fewer substitutions enabled across triplets instead of single sites. Likewise, the incongruencies over the reconstructed topologies were minor, similar to the previous analyses (Fig. 1e). When branch lengths were examined, the average ranking of the criteria presented a different trend compared to c_0-c_2 , whereby BIC and DT were inferior to AIC and AICc (Table 4, last column; Supplementary Fig. S3d, and Supplementary Table S3). Nonetheless, the differences among them were still minute.”

Minor comments

>Over the last 50 years, a plethora of evolutionary models has been developed, each implying different hypotheses on the pattern of nucleotide evolution.

I don't think you want to say that models imply **hypotheses****, rather than represent different assumptions about how we think evolution operates and what is important to model.**

Done. This sentence was rephrased as follows:

“Over the last 50 years, a plethora of evolutionary models has been developed, each relying on a different set of assumptions regarding the dynamics of nucleotide evolution.”

> These, and other extensions such as accounting for the GC content, sum up to an excessive number of possible substitution schemes

The number is not **that**** excessive; clearly we don't need that many models, but there are only 203 unique reversible matrices x small fixed number of frequency estimators x small fixed number of commonly used rate variation distributions.**

Following this and other comments, we revised the introduction and omitted this sentence.

> The increasing number of parameters grants the model the strength and flexibility to fit different datasets and to capture their complexity

I don't think the adjective **strong**** is something you can apply to a model.**

We rephrased this sentence as follows:

“Accounting for more parameters grants a model the flexibility to fit different datasets and capture their complexity.”

> When the sample size is small compared to the number of parameters, it is advised to use the corrected version of AIC, termed AICc^{21,22}, since the former is only valid asymptotically as the size of the data approaches infinity

AIC-c is also only "asymptotically" valid (both AIC and AIC-c use a version of the central limit theorem for their derivation)

As a result of shortening the introduction, this part of the Introduction was removed.

> First, we observed that each of the six criteria selected models that managed to recover the topology of the true tree in 50-51% of the datasets (Table 2)

There's something not quite right with the grammar in this sentence.

We understand the confusion caused by our phrasing and rephrased it:

“when reconstruction the true topology was examined, the six criteria performed similarly as they all selected models that correctly recovered the topology of the true tree in 50 to 51% of the datasets (Table 2, first column).”

> excluding alignments that contained less than four sequences, less than 100 alignment sites

****Fewer****, not ****less****, please.

Corrected. Thank you.

> To allow for comparison across trees with different number of tips, the RF distance of each dataset was divided by the total number of splits

Why do you need to compare trees with different numbers of tips?

Initially, in order to perform a paired t-test and present the average distance across all datasets, we had to make the distances over large trees comparable to the distances over small ones. Therefore, we divided the distances by the number of tips. Yet, this procedure biased the results and obscured the actual magnitude of the relative distances (the performances over small datasets received higher weights due to the division in very small numbers). To overcome these limitations and the potential consequences

of normalizing the data, in the revised manuscript we present the result using two alternative strategies. First, we report the average ranking of the criteria across all datasets rather than the average distance, and compute the statistical significance using the pairwise Wilcoxon signed rank tests adjusted for ties with the Bonferroni correction for multiple testing. Second, we averaged the distances over subsets of the data, binned according to tree size. This makes the scaling procedure unnecessary, and further enables readers to appreciate the magnitudes of the distances across increasing tree sizes.

Reviewer #3

In this manuscript, the authors examine the practice of model selection prior to phylogenetic inference. Their main questions concern comparisons of the various criteria used for model selection (e.g., AIC, AICc, BIC, etc.) and the impact of the criterion used on the accuracy of the inferred phylogeny. One of their main findings is that there is generally disagreement among criteria in selecting the model, but that the criteria all perform similarly in terms of the accuracy of the inferred phylogeny. They also consider the estimation of branch lengths and the inference of the ancestral (root) sequence. Again, differences in inference accuracy among the criteria were small, though in this case there were statistically significant differences among the models selected in terms of branch length accuracy. Finally, the authors compare use of a criterion for model selection to the specification of a complex model (i.e., GTR+I+G) without model selection, and find that the specification of GTR+I+G does not lead to a loss of accuracy in the inferred tree, though it may have an impact on the estimation of branch lengths. The authors thus recommend that the practice of model selection be abandoned in favor of a priori specification of a sufficiently complex model, saving the computational effort required in the model selection stage of the inference procedure.

The paper is well-written overall, and makes some important points, with which I agree in many cases. However, I think the significance and impact on phylogenetic practice may be overstated. For example, the authors state several times that they find it surprising that a consensus has not arisen in the literature with regard to which criterion should be used for model selection. But I think that a very plausible explanation for this is that empiricists are well-aware of the main finding of this paper — namely, that it's important to get approximately the correct model (e.g., we wouldn't want to use JC if the data arose from GTR+I+G, as the authors show), but once we get approximately the correct model, the specific choice isn't very important, and thus the criterion used for model selection is not crucial.

We thank the reviewer for the positive and constructive feedback. We agree that the importance of which criterion to use for model selection was somewhat overstated. However, this topic is widely debated in research (as indicated in the many studies we cite) and we believe that either this should be resolved or explicitly established. To reduce our emphasis on this matter, we revised the manuscript in several locations throughout the manuscript.

Second, it isn't really surprising (and definitely not "remarkable", as the authors state on line 263) that GTR+I+G works well in many cases. Since the parameters of the GTR+I+G model will be estimated during inference of the phylogeny and since the simpler models are nested within this one, if a simpler model was used to generate the data, we'd expect the parameter estimates to be similar to those

under for the simpler model. So I agree completely that this is a reasonable thing to do, and this study bears this out.

Following the comments raised in this review, we rephrased the text to give more emphasis to the fact that JC was only slightly inferior to the other strategies. In the Results, pages 12-13 we write:

“Specifically, we employed GTR+I+G and JC, representing the most complex and simplest models. We also compared performance when using the model that was used to simulate the data (i.e., the true model). While our initial hypothesis was that a model selection step would prove beneficial, our results pointed to the contrary. The percentage of correctly inferred tree topologies was highest for the GTR+I+G model across all simulation sets, slightly better than that achieved for all other model selection criteria, and in particular better than under the true model (Table 2). Corroborating this observation, a similar trend emerged when the RF distances were examined, especially for simulation sets c_1 - c_3 (Table 3). Peculiarly, inference with the oversimplified model JC resulted in a reduction of only ~2% in the number of correctly inferred tree topologies compared to the other strategies, although analysis of the RF distances demonstrated that this decreased performance was statistically significant (Tables 2 and 3; *p-value* < 0.05 when comparing JC to all other strategies across all simulation sets; pairwise Wilcoxon tests following the Bonferroni correction). This suggests that the most appropriate model is not of major importance for topology reconstruction, yet, the introduction of additional parameters may be beneficial.”

We also discuss this point in page 17:

“While different model selection criteria differ in their chosen model, they select features of models that the data seem to support (Fig. 3). It is reasonable that the most parameter-rich model, which combines all of these components, would lead to similar inferences in the risk of including more noise. This raises the question whether any model could suffice. It has been previously shown that using an oversimplified model when the assumed evolutionary patterns are known to be violated deteriorates the accuracy of inference, and in such cases, complex models should be used⁵⁸⁻⁶¹. Surprisingly, in our analysis the recovery rate of the true tree topology by JC was only ~2% lower than the rates obtained with the various model selection criteria, and this gap decreased for the more complex simulation sets (Table 2). The marginally inferior performance of JC is not specifically attributed to small or large trees, but is quite constant across all tree sizes (Supplementary Fig. S2 and Supplementary Table S2). These findings suggest that in many cases there are no major differences among the alternative models, and that any model can serve just as well.”

I also have a few minor comments and wording issues, listed below by line number:

— line 29, “all criteria lead to similar inferences” — I think the authors need to be specific here. Upon reading the paper, we can see that this is certainly true for inference of the phylogeny, probably true for inference of the ancestral sequence, and possibly not true for inference of branch lengths. But there

are other things we might also want to infer. For example, inference of changes in base frequency composition or transition/transversion bias may also be of interest. The authors later mention such possibilities.

Done. We rephrased this part of the abstract as follows:

“We demonstrate that although incongruency regarding the selected model is frequent, all criteria lead to very similar inferences. When topologies and ancestral sequence reconstruction are the desired output, choosing one criterion over another is not crucial. However, for the assessment of branch lengths, the relative performance of the criteria differs between different simulation scenarios. Moreover, we show that for the former tasks, skipping the model selection step and using instead the most parameter-rich model leads to similar inferences, thus rendering this time-consuming step nonessential, at least under current strategies of model selection.”

In addition, we elaborate more on inferences where model selection might be beneficial in the Discussion (pages 16-17):

“It should be noted that in this work only a specific set of commonly used nucleotide substitution models were studied while the effect of other nucleotide substitution models as well as choosing among amino acid matrices and different codon models remains to be studied. Importantly, in some applications the benefit of using model selection is evident, e.g., when transition-transversion and GC-content biases are of interest⁷ or for the inference of positive selection⁵⁷. The main difference between the mentioned inferences, i.e., those that are robust to model selection versus those that might not be, is that in the latter the model selection is inherently important for the inference task, while in the former the substitution model can be regarded as a nuisance parameter. “

— line 52, “excessive” might be too strong — might not evolution happen in a large number of different ways?

Following this and other comment raised by the other reviewers, the Introduction was revised and this sentence was omitted.

— In the paragraph that starts at line 63, I think that likelihood-based methods like AIC, BIC, etc., are referred to twice, at different places in the paragraph.

In this revised submission, we shortened the Introduction and thus the double references were also removed.

— line 91, the statement “reflects the statistical power of the comparison” is too vague; something more precise, for example, “the magnitude of the BF quantifies the relative strength of evidence for the two models”, is needed.

We agree. We rephrased this sentence:

“The magnitude of the Bayes factor (BF), namely, the ratio of the marginal likelihoods of two models, quantifies the strength of evidence that one model is more appropriate to describe the data than the other¹⁹.”

— line 92, “since marginal likelihood is not a closed form expression ...” — but in general there may be a closed form for the marginal likelihood. This doesn’t happen for problems in phylogenetics, but this statement makes it sound like that’s generally true. Much of the remainder of this paragraph should be re-worded to be more precise, differentiating the phylogenetic setting from the basic statistical principles.

We agree. Notably, the main problem is that exploring the entire parameter space is not feasible. In the revised version, this paragraph was rephrased as follows (page 4):

“Notably, handling the uncertainty within model testing by the ML criteria depicted above is accomplished by accounting for the number of parameters assessed in the computation, but not for the type of processes they represent. For example, the penalty for a parameter that distinguishes between transition and transversion would be identical to the penalty imposed for a parameter that assesses the number of invariant sites. In contrast, under the Bayesian approach, model selection can be performed using the marginal likelihood, which is the probability of the data given the model, while marginalizing the estimates (Table 1). The magnitude of the Bayes factor (BF), namely, the ratio of the marginal likelihoods of two models, quantifies the strength of evidence that one model is more appropriate to describe the data than the other¹⁹. Since the marginal likelihood for phylogenetic interpretation consists of high dimensionality and the wide range of values cannot be enumerated, its computation is not always feasible. Several methods that estimate the Bayes factor or the marginal likelihood for model selection in phylogenetic analyses have been proposed, with variable tradeoff between computation times and accuracy^{20–27}.”

— line 141, “to sort” -> “to understand”

Fixed.

— **line 141, remove comma after “criteria”**

Fixed.

— **line 256, remove the word “well”**

Fixed.

— **line 256, “evident” -> “evidenced”**

Fixed. We changed it to “implied”.

“Model selection is considered as a fundamental step in the process of phylogeny reconstruction and has penetrated into the broad phylogenetic community, as implied by the ubiquitous use of the existing tools for model selection.”

— **line 283, “branch lengths estimations” -> “branch length estimates”**

Fixed.

— **line 289, “nucleotide models” -> “nucleotide substitution models”**

Fixed.

— **line 301, “among genomes” is maybe too broad a statement — the paper doesn’t really deal with genome-scale settings**

We agree. We replaced “genomes” with “organisms”:

“To conclude, our results imply that model selection may be unnecessary when one is interested in inferring ancestral sequences or in revealing the phylogenetic relationships among genes and organisms.”

— **Table 1, remove the extra “for” in the description of the BF**

Corrected.

Reviewers' Comments:

Reviewer #1:

Remarks to the Author:

Review by Jeff Thorne of "Is model selection a mandatory step for phylogeny reconstruction", a revised manuscript submitted by Abadi et al. to Nature Communications

This revised version carefully addresses all of the concerns that I had raised regarding the earlier version. This is a nice work.

Reviewer #3:

Remarks to the Author:

The authors have done a careful and thorough job of responding to the comments I raised in my initial review. I am satisfied that the manuscript is now suitable for publication.